

# psit 1.0: A System to Compress Lagrangian Flows

Alexander Pietak[1], Langwen Huang[2], Luigi Fusco[2,3], Michael Sprenger[4,*], Sebastian Schemm[4,*], and Torsten Hoefler[2]

[1]Department of Mathematics, ETH Zürich, Zurich, Switzerland
[2]Scalable Parallel Computing Lab, ETH Zürich, Zurich, Switzerland
[3]Microsoft Ibérica, Madrid, Spain
[4]Department of Environmental Systems Science, ETH Zürich, Zurich, Switzerland
[*]These authors contributed equally to this work.

**Correspondence:** Alexander Pietak (apietak@ethz.ch)

**Abstract.** Meteorological simulations produce large amounts of data, which can represent a challenge when trying to store, share, and analyze it. As weather and climate models increasingly simulate the atmosphere at higher spatio-temporal resolution, it becomes imperative to compress the data effectively. While compression algorithms exist for weather data stored in a gridded Eulerian frame, there are, to date, no specialized alternatives for data stored in the Lagrangian frame. In this study, we present

psit, a system to compress weather data stored in the Lagrangian frame. The system works by mapping the trajectories to a grid structure, performing additional encodings on these, and passing them to either the `JPEG 2000` image compression algorithm or `SZ3`. The specialty of the algorithm is the mapping phase and the following encodings, which generate the grids in a way that allows the aforementioned compression algorithms to perform well. To gauge the performance of psit, we test a variety of metrics. We demonstrate that in the majority of cases, equivalent or superior compression performance is attained through the

utilization of psit as opposed to naive compression with `ZFP`. We also compare compression with measurement inaccuracies. Here, we show that the density of $168$ hour long trajectories compressed with a ratio in the range of $30$ to $40$ behaves similarly to trajectories calculated from uncompressed wind fields with additional random perturbations with magnitude of $0.1\,\mathrm{m\,s^{-1}}$ in the horizontal and around $6 \cdot 10^{-3}\,\mathrm{Pa\,s^{-1}}$ in the vertical component. Additionally, we conduct two case studies in which we discuss the impact of compression on the study of warm conveyor belts associated with extratropical cyclones and the impact

of compression on the radioactive plume prediction of the Fukushima incident in 2011.

## 1 Introduction

With advancements in parallel computing and the introduction of larger and more powerful HPC systems, the size and complexity of today's meteorological models have increased significantly. Such an increase in complexity leads to a rise in the amount of output data, reaching the range of dozens of petabytes (Hoefler et al., 2024, 2023). Storing and analysing such

a massive amount of data poses significant challenges. Consequently, different approaches have been developed to address this issue. One approach works by reducing the number of elements that are stored (Tintó Prims et al., 2024). In the case of weather data, this can be achieved, for example, by reducing the grid resolution of the stored data or decreasing the temporal interval at which data is stored. This approach is already in use for the ERA5 weather dataset (Tintó Prims et al., 2024). A





second approach that is commonly used focuses not on reducing the number of elements but on decreasing the space occupied
by each of them. This is the field of compression algorithms, which reduce data size by exploiting structure present in the
underlying data, a technique commonly used in multimedia, as seen in formats like JPEG (Wallace, 1991) and H.265 (Sullivan
et al., 2012). For the compression of scientific data, we usually prefer lossy compression, as, due to the inherent noise present,
lossless compression algorithms will not be able to achieve large compression ratios (in the range of 2 to 4 times). We will
therefore focus on lossy compression techniques.

Several lossy compression algorithms for the compression of multidimensional floating-point data have been developed.
A (non exhaustive) list includes: `SZ3` (Liang et al., 2023; Zhao et al., 2021; Liang et al., 2018), `ZFP` (Lindstrom, 2014;
Diffenderfer et al., 2019), and THRESH (Ballester-Ripoll et al., 2019; Ballester-Ripoll and Pajarola, 2015). Previous research
by Tintó Prims et al. (2024) has demonstrated the effectiveness of these floating-point compression algorithms in compressing
weather data in the Eulerian frame. There are also specialized compression algorithms developed specifically for weather data,
such as VAEformer (Han et al., 2024), an autoencoder- based compression algorithm designed for the compression of the ERA5
weather dataset. Hence, a wide variety of compression algorithms exist that can be utilized for the compression of weather data
in the Eulerian frame.

On the other hand, meteorologists not only work with the Eulerian representation but also often consider a Lagrangian one.
In this representation, we do not have a multidimensional grid but instead track the position and properties, like temperature,
of small air parcels carried by the wind, resulting in so called trajectories. This representation is crucial for certain types of
analysis, such as identifying warm conveyor belts (Wernli and Davies, 1997b; Joos and Wernli, 2012; Schemm et al., 2013)
or stratosphere-troposphere exchanges (Holton et al., 1995; Škerlak et al., 2014). However, here we also face the problem
of an increase in output data and therefore seek to compress it. Compared to multidimensional grid data, the landscape of
compression algorithms for weather data trajectories is much poorer. Existing compression algorithms for trajectory data
mostly focus on data generated from GPS tracking of vessels or, more generally, the movement of objects in space over many
time steps (Makris et al., 2021). Examples of such algorithms include Dead-Reckoning, Douglas-Peuker, and Time-Ratio
(Makris et al., 2021), all of which reduce data volume by strategically removing individual time steps while preserving the
general trajectories. However, these methods are not suitable for compressing trajectories originating from weather data for
two main reasons. First, the number of time steps in weather trajectories is typically much lower than in GPS trajectories. For
GPS trajectories, the number of time steps might be in the $10'000$ range, while for weather trajectories, depending on the use
case, one might work with only a few dozen (e.g., hourly to six hourly temporal resolution). A compression algorithm that
reduces the number of saved time steps will not be effective when the number of time steps is already very low. Additionally,
removing time steps from weather trajectory data may not be permissible due to the specific requirements of subsequent
research. Secondly, weather trajectory data often includes additional variables, such as temperature or potential vorticity, that
are of interest along the trajectories. The above mentioned trajectory compression algorithms do not account for these additional
variables, only for the position, which makes them unsuitable for compressing such weather data. For these reasons, specialized
compression algorithms for trajectories originating from weather data need to be considered.





To our knowledge, no readily available compression algorithm exists for trajectory data emerging from weather data, and research in this area appears to be limited. Some research has been conducted on the compression of data on unstructured grids, for example, the paper by Iverson et al. (2012), which is also not applicable to our use case. To address this gap, we present psit, a lossy compression method for Lagrangian flow data. Psit works by transforming the trajectory data into 2D grids and passing those grids to existing compression algorithms, specifically `JPEG 2000` (Skodras et al., 2001) and `SZ3`, where `JPEG 2000` has shown its potential in internal research and is used by the visualization community due to its adaptive scalability property (Woodring et al., 2011). This approach leverages the refined performance of these compression algorithms to create a compression pipeline specifically designed for trajectory weather data.

## 2 Method

The general idea behind psit is to take the trajectories and intelligently transform them so that they can be passed to existing compression algorithms. This general pipeline is illustrated in Fig. 1. The input expected by the dense data compressors (`JPEG 2000` and `SZ3`) consists of 2D arrays (i.e., grids), which ideally should be smooth, i.e., the difference between neighboring entries of the array should be small. Note that a single 2D array (representing one channel) can be passed to the `JPEG 2000` algorithm, as it is able to handle an arbitrary number of channels (individual 2D arrays) and is not limited to the default three color channels (red, green, and blue) used for images. Therefore, in the first step (the mapping phase), the trajectories are transformed into such grids. During this mapping, it is crucial to assure the resulting grids smoothness, as greater smoothness allows both compression algorithms to perform more effectively. Once the mapping step is completed, encoding schemes can be applied to them, namely delta and color encoding. These encoding schemes modify the grids in a way that leads to better compression performance for the previously mentioned compression algorithms. In the next step, the encoded grids are passed to a dense data compressors to perform the compression, and the compressed data is subsequently stored to disk. These four steps form the basis of psit and are discussed in more detail in the rest of this section.

The decompression phase mirrors the compression process: the data is loaded from disk and then decompressed using the same algorithm employed during compression (e.g., `JPEG 2000`). After this, the encoded grids are decoded (delta and color decoding) to obtain the mapped 2D grids. Finally, these 2D grids are transformed back into trajectories using an inverse mapping of the original one. After this step, the decompression is complete, and the trajectory data should closely match the original data, with only the errors introduced by the compression algorithm differentiating them.

### 2.1 Input data

In the following, we will provide a definition of what a trajectory and a possible input file are in order to establish a common basis for later discussions. We define a trajectory as a sequence of positional variables (longitude, latitude, and pressure, which



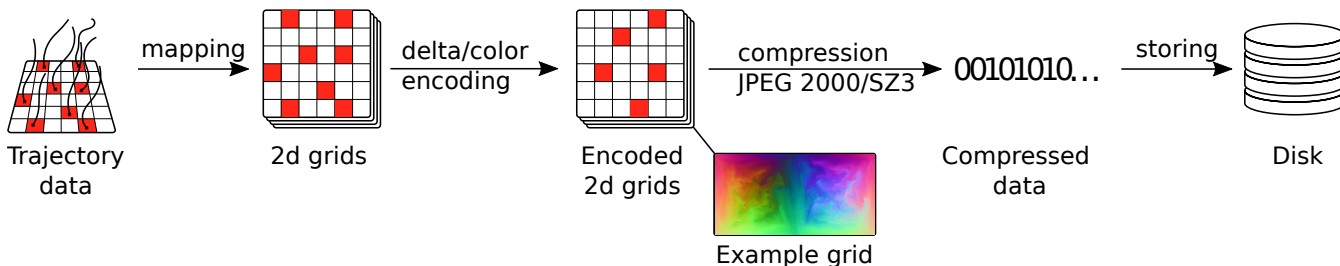

**Figure 1.** The compression pipeline of psit. Trajectory data is converted into grid data, to which then delta or color encoding can be applied, after which it is compressed and stored to disk. The decompression works the same in the opposite direction.

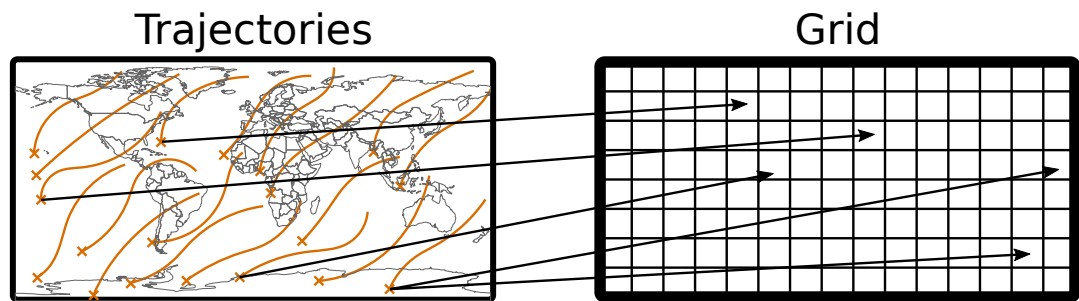

**Figure 2.** Visual representation of a mapping, each trajectory is assigned to some grid points, note that for visual clarity only some mappings are drawn. The way in which this association is done forms an integral part of our pipeline and is based on solving a minimal weight full bipartite matching problem.

indicates the height) defining a 3D path on Earth, along with additional sequences that store variables like temperature or humidity along it. Each entry in these sequences represents one time step. Multiple such trajectories (in the range of a few million) can then be collected together to form a trajectory file, which would function as input to our compression algorithm.

For example, an input file might consist of one million trajectories spanning 13 time steps, with each trajectory also storing temperature and humidity. At the first time step, the trajectories are uniformly initialized over 26 pressure levels spanning the entire globe, with some predefined distance between the different starting points. This would result in a file storing a total of five million individual sequences (five per trajectory: 3 positional and 2 data variables), where each sequence stores 13 elements.

## 2.2   Mapping

The mapping phase is the first step in the compression pipeline and focuses on transforming the trajectories into grids while ensuring that the resulting grids are smooth. To achieve this, a method for storing the trajectory data as grids must be established. A simple approach is used: each trajectory is associated with at least one grid point, as visually represented in Fig. 2. The next step consists of translating the data from the trajectories into grids. For this, we first create an individual grid for each time



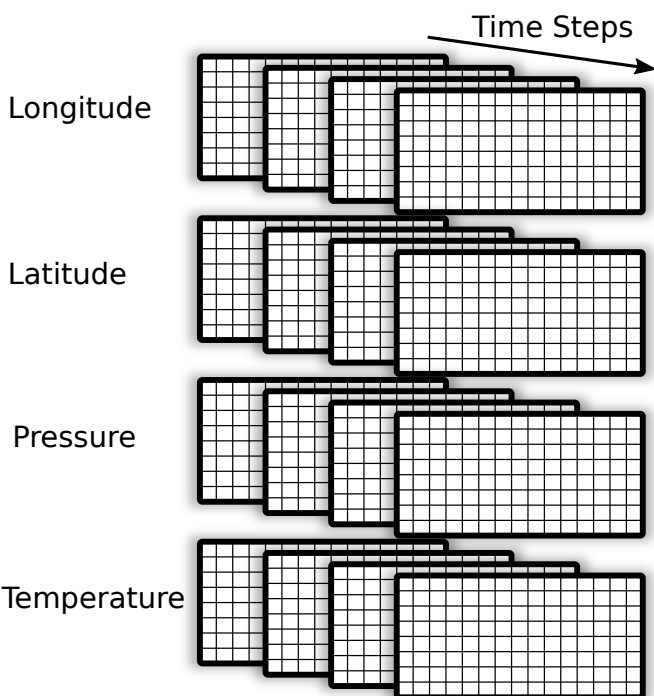

**Figure 3.** Visual representation of the splitting of the trajectories onto multiple grids. For each time step and data variable an individual grid is created which stores the data of the trajectories of that data variable for one specific time step. Shown are four grids for each variable.

step, and because a trajectory consists of multiple data variables (3 positional and an arbitrary number of additional ones), we

also create an individual grid for each of those (Fig. 3). After all these grids are created, we simply set the values of the grid points to the value of the associated trajectories data variable at that specific time point. This way, a collection of 2D grids can be created that store the same information as the trajectories but are inherently 2D. For example, the previously introduced trajectory file consists of one million, 13 time step long trajectories, which also store temperature and humidity. This file would be converted into 65 individual grids, where each grid would have at least one million entries, e.g., a dimension of $1500 \times 750$.

While creating such an initial association can be done trivially, there are constraints that make the task more difficult. The first and most crucial constraint is that the resulting grids need to be smooth. Secondly, we must ensure that every trajectory maps to at least one grid point. If this is not the case, a trajectory would have no representation in the compressed data and would therefore be lost during compression. These constraints, especially the smoothness one, prevent the use of naive mapping methods and require the use of more sophisticated techniques.

In our research, we analysed different mapping methods. The one that yields the best results is based on solving a minimal weight full bipartite matching problem, originating in graph theory (Burkard et al., 2012). A second promising approach is based on solving a linear program (LP) (Bertsimas and Tsitsiklis, 1998). While this approach is conceptually sound, it quickly



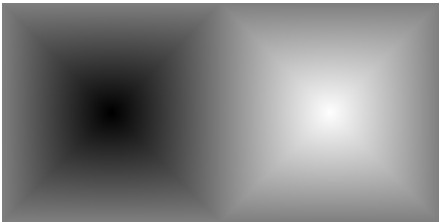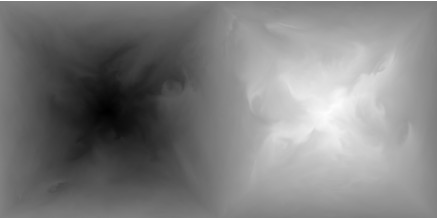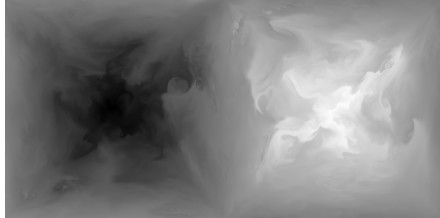

**Figure 4.** Latitude grids with 550hPa starting pressure created with the bipartite mapping method. Three different time steps are displayed the leftmost image is at the first time step, the second one is 12 hours later and the rightmost one is 24 hours later.

becomes computationally infeasible, so it found little use in the finished pipeline. However, we still include it here for its theoretical insights.

### 2.2.1 Bipartite Mapping

For this mapping method (results in Fig. 4, with an example in Fig. 5), the problem is solved using concepts from graph theory, more specifically by solving a minimal weight full bipartite matching problem. Hence, following this approach, we have two sets of nodes: one called "workers" and the other called "tasks", with weighted edges between them constraining which workers may be assigned to which tasks and what the cost of this assignment is. The goal is to match each worker to exactly one task while trying to keep the overall cost of the matchings as low as possible.

This minimum weight full bipartite matching problem can be easily applied to our use case. Namely, the workers become the trajectories and the tasks the grid points. The weights are then defined as the Euclidean distance between the trajectories and the grid points at the first time step after both are translated into 3D Cartesian space. To transform the grid points into Cartesian space, a projection onto a 3D sphere is used. This projection method should lead to the grid points being homogeneously distributed over the globe, making a simple longitude-latitude mapping unsuitable. We chose the projection presented in Section 4 of the paper by Calhoun et al. (2008). The trajectories are transformed into Cartesian space by taking their spherical coordinates of longitude, latitude, and pressure and converting them to Cartesian $x$, $y$, and $z$ coordinates. This transformation leads to the creation of a spherical shell, representing a mismatch in dimensionality between the grid points and the trajectories. Therefore, the spherical shell is binned into discrete levels over its pressure, resulting in the creation of multiple spheres for each of the bins. This means that, in addition to a grid being created for each time step and data variable, individual grids are also created for each of these pressure levels. If the algorithm then minimizes the Euclidean distance between the trajectories and the grid points, the resulting grids should become smooth, as trajectories with similar starting positions should be mapped to grid points that are close to one another, and we have that trajectories starting at similar spatial positions behave rather similarly over time (hence locating them next to each other results in a smoother grid). So, by representing our mapping problem as such a graph problem and solving it, we should end up with a valid and smooth mapping.



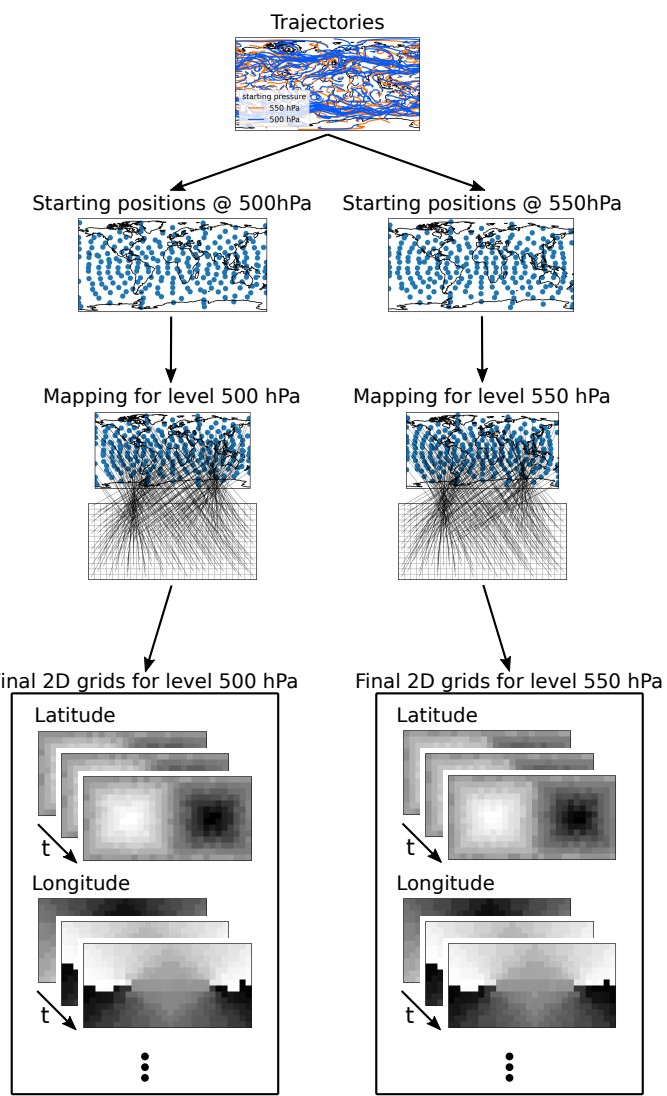

**Figure 5.** Example of how the bipartite mapping works. For this 464 trajectories starting at two different pressure levels are considered.





One problem with this mapping via minimal weight full bipartite matching is that if the number of trajectories is less than the number of grid points some grid points will not be mapped, leading to holes in the resulting grid. While it theoretically is possible to have the same number of trajectories as grid points, tests showed that this can become computationally infeasible and that in practice we need more grid points than trajectories. Therefore, we need to devise a way to handle this. The hole

filling works by taking all the grid points that do not get mapped and mapping their closest trajectory (Euclidean distance in Cartesian space) to them (note that this means that a trajectory may map to multiple grid points at the same time). By using this method, the holes can be filled in an easy manner that does not require any costly computations.

### 2.2.2 LP Mapping

We also considered using linear programming. While this approach has high theoretical appeal, its practical usability is limited

by the fact that the resulting LP becomes very large and requires a lot of resources to solve, while delivering nearly identical results to the approach based on minimal weight full bipartite matching. For our use case, we will model the mapping problem as an integer linear program (ILP), for which we then show that its LP formulation is integral, i.e., solving the LP formulation instead of the ILP one delivers the same results. This allows us to use a much cheaper LP solver, resulting in a large decrease in computational resources. In order to define an integer linear program, we need to specify an objective and a set of constraints

that create a valid and smooth mapping from trajectories to grid points.

We first start with an intuitive definition of the objective and the constraints. The objective function is designed to minimize the sum of distances between the trajectories and their associated grid points. For the constraints, we say that each trajectory must map to at least one grid point and that each grid point must be mapped to. By the property of spatial continuity, a minimization of the objective function should lead to a smooth grid, while the constraints ensure that the mapping is valid. The

155 rigorous mathematical definition is then given by:

Let the binary variables $x_{t,p} \in \{0,1\}$ denote if a trajectory $t \in T$ is mapped to grid point $p \in P$, $0$ if not mapped, $1$ if mapped. And let the variable $d_{t,p} \in \mathbb{R}_{\geq 0}$ denote the Euclidian distance in 3D Cartesian space (defined analogue to the bipartite mapping method) between a trajectory $t \in T$ and a grid point $p \in P$. The resulting ILP is then given in Eq. (1):

$$\text{minimize} \sum_{(t,p) \in T \times P} d_{t,p}\, x_{t,p} \tag{1}$$

$$\text{s.t.} \sum_{t \in T} x_{t,p} = 1 \quad \forall p \in P \tag{2}$$

$$\sum_{p \in P} x_{t,p} \geq 1 \quad \forall t \in T$$

While the variables $x_{t,p}$ are defined to be binary, making the problem an *integer* LP, we can prove (see Appendix A) that the LP formulation, achieved by defining $x_{t,p} \in [0,1]$, is integral. This means that instead of an ILP solver a much cheaper LP solver can be used during calculation.





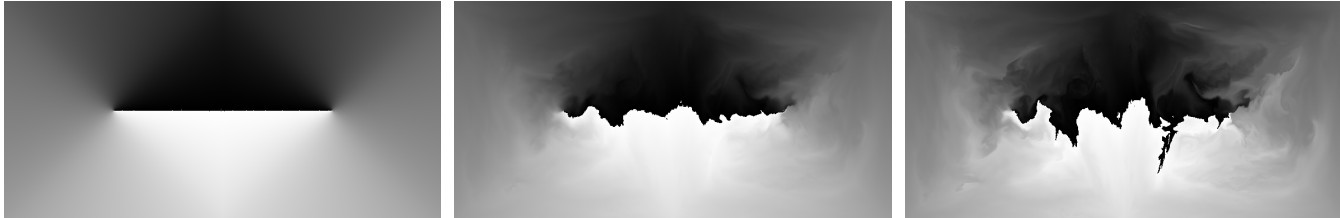

**Figure 6.** Longitude grids with 550hPa starting displayed over different time steps (0 h, 12 h, 24 h). At the date line there is a discontinuity which starts to warp over time.

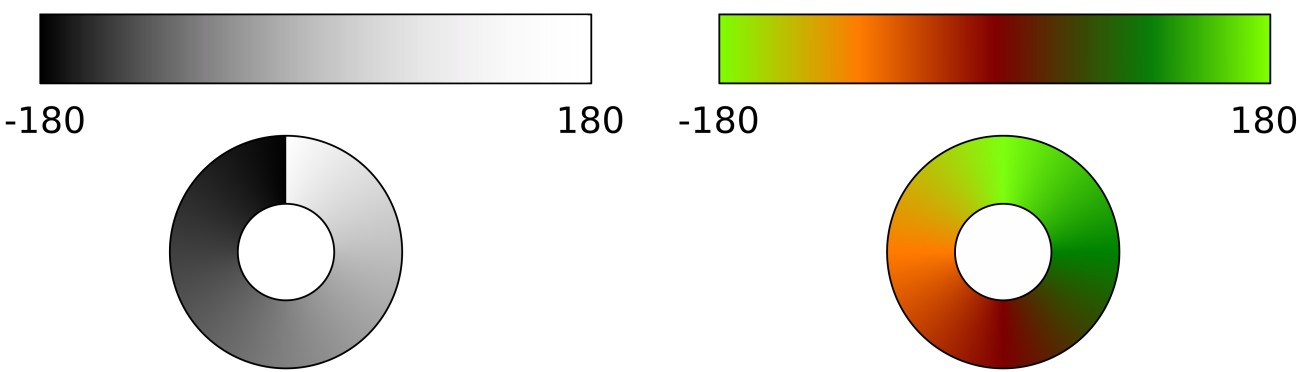

**Figure 7.** Qualitatively demonstration of the discontinuity line at the date line. On the left only a singular variable (black to white) is used to represent the longitude, a discontinuity is created. On the right two variables and a cosine sine mapping is used, displayed as red and green. Now it is possible to create a continuous color cycle and no discontinuity is present.

## 2.3 Color Encoding

To motivate color encoding, we must examine the longitude grids created by the previously explored mappings (Fig. 6), which mapping we look at (LP or bipartite) is not important as they deliver very similar results. As we can see from Fig. 6, there is a discontinuity line around the position of the dateline at longitude $180°$, which begins to warp as time progresses. This discontinuity arises because trajectories with a longitude value of $-180$ (black pixels) are adjacent to trajectories with longitude values of $180$ (white pixels). Such jumps in pixel values lead to worsened compression performance, as shown by experiments where we observed a decrease in RMSE error in the longitude variable by a factor of $1.4$ when compressing with color encoding compared to no color encoding (using a compression factor of 15).

The way we eliminate the discontinuity line is by using multiple variables instead of one, which maps the $[-180, 180]$ longitude range to the black to white pixels. Hence, we map the $[-180, 180]$ to a multidimensional space. If we use two or three variables, this multidimensional space can be represented by a color bar. This concept is illustrated in Fig. 7, where on





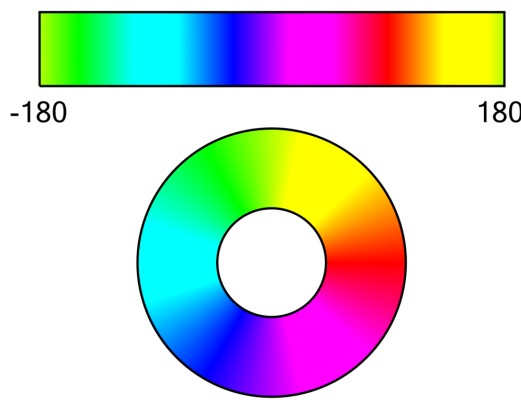

**Figure 8.** Color cycle produced by HSV mapping, with constant value of 1 for saturation and value and varying hue.

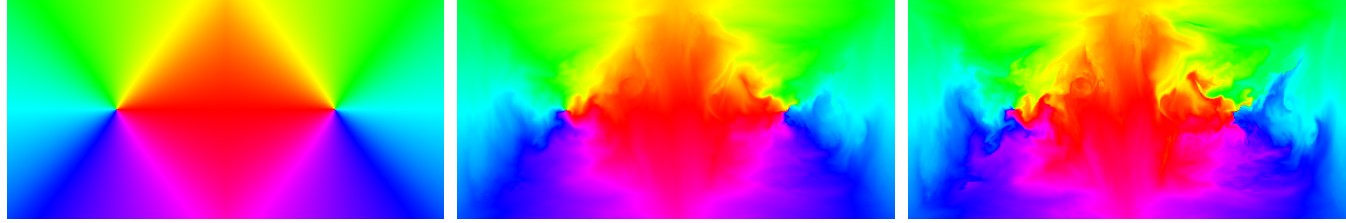

**Figure 9.** HSV color encoded longitude grids with 550hPa starting displayed over different time steps (0 h, 12 h, 24 h). The date line discontinuity of Fig. 6 is gone.

the left, we have a single variable mapping the longitude from black to white, generating a discontinuity in the color wheel. On the right, we use two variables (red and green) with a sine-cosine mapping to create a color bar, which results in a continuous color wheel. This simple concept can define a variety of different color encoding methods. In our research, we explored several such methods, and the ones that provided the best performance are based on utilizing three variables, covering the entire RGB

range. We call them HSV and XYZ color encoding and they are explained in the following sections.

### 2.3.1 HSV color encoding

The HSV color encoding method (Fig. 9) uses the HSV (Smith, 1978) color representation, which, like RGB, is a way to describe color. However, instead of using values for red, green, and blue, it uses hue, saturation, and value to represent color. For our use case, we set the saturation and value to 1 and only vary the hue. For constant saturation and value, a varying hue

generates a continuous color wheel, as shown in Fig. 8. Another reason for setting saturation and value to 1 is that the `JPEG`







**Figure 10.** XYZ color encoded longitude/latitude grids with 550hPa starting displayed over different time steps (0 h, 12 h, 24 h). The date line discontinuity of Fig. 6 is gone.

2000 algorithm might handle dark or washed-out colors differently, as the human eye cannot distinguish them as precisely as strong, bright colors.

The function $f(x)$ for converting a longitude value in the normalized range $[0, 1]$ to an RGB value using an HSV representation is given in Eq. (3). The inverse mapping $f_{\text{inv}}([r, g, b]^T)$, which maps an RGB color value to a hue angle, is provided in Eq. (4).

$$f(x) = \begin{bmatrix} h(5) \\ h(3) \\ h(1) \end{bmatrix} \tag{3}$$

with

$$h(l) = 1 - \max\left\{0, \min\left\{k, 4 - k, 1\right\}\right\}$$
$$k = (l + 6x) \bmod 6$$

$$f_{\text{inv}}([r, g, b]^T) = \begin{cases} 0, & \text{if } c = 0 \\ \frac{1}{6} \cdot \left(\frac{g-b}{c} \bmod 6\right), & \text{if } v = r \\ \frac{1}{6} \cdot \left(\frac{b-r}{c} + 2\right), & \text{if } v = g \\ \frac{1}{6} \cdot \left(\frac{r-g}{c} + 4\right), & \text{if } v = b \end{cases} \tag{4}$$

$$v = \max\{r, g, b\}$$
$$k = \min\{r, g, b\}$$
$$c = v - k$$





### 2.3.2   XYZ color encoding

The XYZ color encoding (Fig. 10) method considers both the longitude and latitude data variables, combining them into a single three-channel color grid. To this end, the longitude and latitude values are taken and converted into Cartesian coordinates, from which the $r$, $g$, $b$ color channels are created from the $x$, $y$, $z$ positional coordinates.

The function $f([x_{\text{lon}}, x_{\text{lat}}]^T)$, which maps both longitude and latitude in the normalized range $[0, 1]$ to RGB values, is given in Eq. (5). The inverse function $f_{\text{inv}}([r, g, b]^T)$, which converts RGB values back to longitude and latitude, is given by Eq. (6).

$$f([x_{\text{lon}}, x_{\text{lat}}]^T) = \frac{1}{2}\left(1 + \begin{bmatrix} \cos\hat{x}_{\text{lat}}\cos\hat{x}_{\text{lon}} \\ \cos\hat{x}_{\text{lat}}\sin\hat{x}_{\text{lon}} \\ \sin\hat{x}_{\text{lat}} \end{bmatrix}\right) \tag{5}$$

where $\hat{x}_{\text{lat}} = \pi(x_{\text{lat}} - \frac{1}{2})$, $\hat{x}_{\text{lon}} = 2\pi(x_{\text{lon}} - \frac{1}{2})$

$$f_{\text{inv}}([r, g, b]^T) = \begin{bmatrix} \frac{1}{2\pi}(\arctan2(\hat{g}, \hat{r}) + \pi) \\ \frac{1}{\pi}(\arctan2(\hat{b}, \sqrt{\hat{r}^2 + \hat{g}^2}) + \frac{\pi}{2}) \end{bmatrix} \tag{6}$$

where $\hat{r} = 2r - 1$, $\hat{g} = 2g - 1$, $\hat{b} = 2b - 1$

## 2.4   Delta Encoding

We also looked into using delta encoding as part of our compression pipeline (Fig. 11). The principle behind delta encoding is that instead of storing time sequenced data directly, the delta between subsequent time steps is calculated and saved. While the calculation of the delta itself will not lead to a decrease in data, it might reduce entropy, which can beneficially impact the performance of compression algorithms. We will formulate this more rigorously in the rest of this section.

We will start with a naive implementation of delta encoding and show that it is not suitable. We will use the following notation: At time step $i$, the full grid is denoted as $g_i$, and the delta between the grids at time steps $i$ and $i - 1$ is denoted as $\Delta_i$. It is calculated by subtracting the previous grid from the current one: $\Delta_i = g_i - g_{i-1}$. Then, instead of passing the full grid sequence $(g_1, g_2, ..., g_N)$ to the dense data compressors, we use the delta sequence $(g_1, \Delta_2, \Delta_3, ..., \Delta_N)$. The problem with this simple delta encoding approach is that the error starts to accumulate over time. To see this, we define

$$g_i' = g_i + \epsilon_i \tag{7}$$

$$\Delta_i' = \Delta_i + \eta_i = g_i - g_{i-1} + \eta_i \tag{8}$$





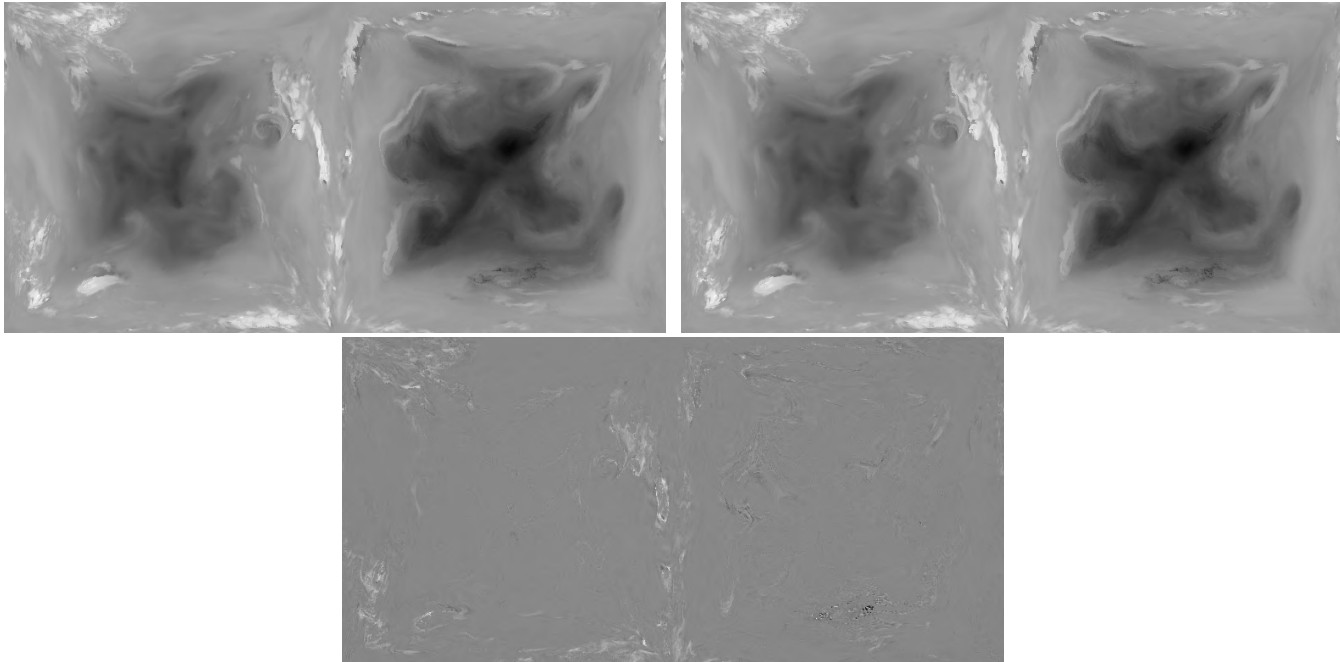

**Figure 11.** Full grids of the potential temperature for 22 h and 24 h with the delta between them.

as the full frames and the delta frames which have gone though compression and therefore have an error $\epsilon_i$ and $\eta_i$ applied to them. The grid at time step $i$ that gets reconstructed by this is:

$$g_i^r = g_1' + \sum_{j=2}^{i} \Delta_j' = g_i + \epsilon_1 + \sum_{j=2}^{i} \eta_j \tag{9}$$

Therefore the delta frame compression errors $\eta_j$ start to accumulate over time. We can solve this problem rather easily. Instead of defining a delta frame from the "perfect" grid $g_{i-1}$, we use the reconstructed $g_{i-1}^{r*}$ grid:

$$\Delta_i'^* = \Delta_i^* + \eta_i = g_i - g_{i-1}^{r*} + \eta_i, \quad g_{i-1}^{r*} = g_1' + \sum_{j=2}^{i-1} \Delta_j'^* \tag{10}$$

The $g_i^{r*}$ can then be reformulated to

$$g_i^{r*} = g_1' + \sum_{j=2}^{i} \Delta_j'^* = g_i + \eta_i \tag{11}$$

As we can see here the error does not start to accumulate over time, therefore this method is to be preferred over the naive one.

## 2.5 Implementation details

For the implementation, we had to take certain shortcuts and make adjustments to the theory in order to remain within reasonable computational limits. The primary simplification we made was to limit the possible number of grid points a trajectory may





map to, for both the bipartite and the LP mapping, to a set of its closest neighbors. For bipartite mapping, this was the closest
200 neighbors, while for the LP mapping, it was the closest 50. However, this introduces the additional constraint of needing
the starting locations of the trajectories to be evenly distributed. If they are not evenly distributed, i.e., if there are regions of
low and high trajectory density, the mapping phase will fail, as we reach a point where we have locally more trajectories than
possible grid points to which they may map. Additionally, we need more grid points than trajectories in practice, this is due to
both considering only the closest neighbors and because the algorithmic implementation we chose to solve the minimal weight
full bipartite matching problem performs much faster when there are more grid points compared to trajectories. In practice, we
have around 50% more grid points than trajectories. These are the main considerations we had to take during implementation in
order to strike a balance between computational performance and compression capabilities. In the future, it might be advisable
to revisit these shortcuts, as removing them could lead to a broader applicable field of psit.

Some other minor implementation details are that we only implemented color encoding in combination with the `JPEG`
`2000` compression algorithm, and that when compressing local rather than global trajectories (i.e., starting positions are
bounded inside a box instead of spanning the entire globe), a longitude-latitude projection is used to map the grid points
into 3D Cartesian space. This is because there is no simple inverse function for the presented projection method. Both of these
differences do not have a large impact on the compression performance of psit, but could still be explored in future research.

## 3 Results

In order to gauge the performance of psit we will make five different experiments:

1. Error metrics and comparison to ZFP in order to compare performance to existing alternatives,

2. error value distribution in order to discern the creation of bias in the error,

3. trajectory density comparison against trajectories produced from perturbed and bitrounded wind fields in order to compare the impact of compression with the impact of data assimilation inaccuracies,

4. two case studies, one about warm conveyor belts and one about the Fukushima disaster, in order to see how psit performs in a more realistic environment,

5. and throughput and memory usage to observe the computational performance.

This should provide the reader with an insight into how psit behaves in different situations.

### 3.1 Input Data

During the following experiments we will use the following trajectory files:





`tra_20200101_00`

    A file produced using Lagranto (Wernli and Davies, 1997a; Sprenger and Wernli, 2015), based on ERA5, starting on 2020-01-01 00:00 and going until 2020-01-02 00:00, with a total of 13 time steps (each two hours long). It consists of $5'291'188$ trajectories, distributed across 26 pressure levels at the first time step with horizontal spacing of $50\,\mathrm{km}$, and vertically ranging from $50\,\mathrm{hPa}$ to $550\,\mathrm{hPa}$ in $20\,\mathrm{hPa}$ increments. The additional data variables it stores are potential temperature (TH) and potential vorticity (PV). The potential vorticity data variable is rescaled to the range of $-30$ to $30$ and contains outliers with very large negative values. The file is $1.3\,\mathrm{GB}$ in size.

`tra_20200101_00_permuted`

    This is the same trajectory file as `tra_20200101_00`, but with the trajectory order randomly permuted. Everything else remains the same. The order of the trajectories in the `tra_20200101_00` dataset follows the grid and pressure levels, which leads to continuity (i.e., neighbouring trajectories are similar) if the trajectories are traversed in order. By permuting them, this continuity is removed.

`tra_20000101_00`

    A trajectory file produced using Lagranto, based on ERA5, starting on 2000-01-01 00:00 and extending 168 hours until 2000-01-08 01:00. It consists of $5'867'016$ trajectories distributed over the default 37 ERA5 pressure levels from $1000\,\mathrm{hPa}$ to $1\,\mathrm{hPa}$, with spacing of $40\,\mathrm{km}$. The additional data variables traced along the trajctories are temperature (T), potential vorticity (PV), and specific humidity (Q). The file is $23\,\mathrm{GB}$ in size.

`tra_20000101_00_permuted`

    The same trajectory file as `tra_20000101_00`, but with the trajectory order randomly permuted. The reasoning behind this is the same as for the `tra_20200101_00_permuted` file.

### 3.2 Performance of psit

    There are different ways in which psit can be configured, leading to different compression behaviors. Here, we present one configuration stack that led to good results with the presented input data files. This configuration is given by the configuration file in Listing 1, for a description of how these configuration files work, please refer to the user manual at (Pietak, 2025a). We are using the `JPEG 2000` compression algorithm combined with delta encoding for every data variable except for pressure, where we use `SZ3` with no delta encoding. Additionally, we utilize the XYZ color encoding scheme. The compression factor, a metric that determines how strong the compression should be, is varied between the different runs and is the same for every data variable except for pressure, where it is $1.5$ times larger. We will use this configuration for all of the following experiments, except for the case study on the Fukushima disaster, as we are working with a different input file there.

#### 3.2.1 Error metrics and comparison with `ZFP`

    In this experiment, a comparison between compression ratio and compression error for both psit and `ZFP` is conducted. The compression pipeline for `ZFP` works by treating each data variable as a 2D array over the time steps and the trajectories and



**Listing 1.** Configuration file for psit used for the runs presented in Section 3.2. The compression ratios are replaced by placeholders `<ratio>` to indicate that they are changed between the different runs, the compression ratio for the pressure variable is 1.5 larger than for the other ones.

```
 1:    method:
 2:        lon: jpeg
 3:        lat: jpeg
 4:        p: sz3
 5:        TH: jpeg
 6:        PV: jpeg
 7:    delta_method:
 8:        lon: r1
 9:        lat: r1
10:        p: none
11:        TH: r1
12:        PV: r1
13:    color_method: xyz
14:    ratio:
15:        lon: <ratio>
16:        lat: <ratio>
17:        p: <1.5*ratio>
18:        TH: <ratio>
19:        PV: <ratio>
20:    bin: 0
21:    color_bits: 16
22:    num_workers: 26
23:    factor: 1.5
24:    exclude: BASEDATE
```

compressing each of them individually. The error norms we use in this study are the L1, RMSE, and L-infinity errors between the original and the compressed/decompressed data, which are obtained by calculating the errors defined by Eq. (12), (13), 295 (14) between each data variable of the original and compressed/decompressed data file.

$$l_1 = \frac{1}{n} \sum_{i=1}^{n} |x_i - y_i| \tag{12}$$

$$l_{\text{rmse}} = \sqrt{\frac{1}{n} \sum_{i=1}^{n} (x_i - y_i)^2} \tag{13}$$

$$l_\infty = \max_{i=1,\dots,n} \{|x_i - y_i|\} \tag{14}$$

where $n$ is the number of data points, $x_i$ the original data and $y_i$ the compressed/decompressed data. As we are using color 300 encoding, we need to handle the longitude and latitude variables differently. Instead of using the difference, we use the central





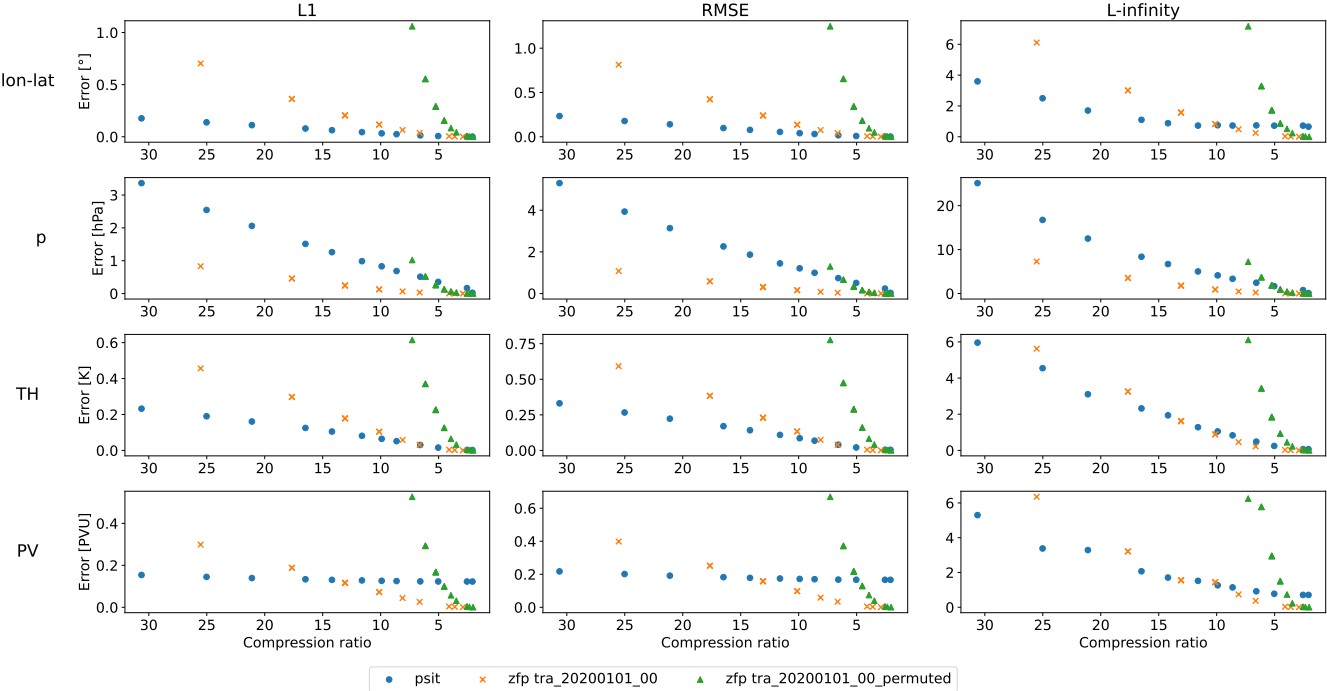

**Figure 12.** Comparison between psit and `ZFP` for the `tra_20200101_00` and `tra_20200101_00_permuted` files. The configuration of our program is described in Subsection 3.2 and the corresponding config file is given in Listing 1. The L1, RMSE and L-infinity error is compared to the achieved compression ratio for the central angle error (lon-lat) and the other data variables (pressure (p), potential temperature (TH), and potential vorticity (PV)).

angle between the input and the output (defined as the angle on a great circle between two points on a sphere). This comparison between psit and `ZFP` provides us with an overview of how psit compares to already existing alternatives and therefore allows us to assess its viability.

In the first run, the `tra_20200101_00` and `tra_20200101_00_permuted` files are considered. The result of this is

displayed in Fig. 12, with the exact values given in Tables B1, B2, and B3. The figure consists of four rows, each with three columns. The rows correspond to the different data variables and the columns to the different error metrics. The units of the errors are the same as those of the original data variable. Note that psit performs the same for the `tra_20200101_00` and `tra_20200101_00_permuted` files, therefore, only one is plotted. The actual compression ratio ranges from 1 to around 30, with a close to linear increase in the error norms. In general, the L-infinity error is larger than the L1 and RMSE errors. For

the central angle, potential temperature, and potential vorticity variables, this relative difference is around a factor of 20, while for the pressure variable, it is around a factor of 6. The difference in relative error between pressure and the other variables is because `JPEG 2000` does not bound the L-infinity norm, while `SZ3` does. Therefore, in applications where the L-infinity error is crucial and some sacrifices can be made in the L1 and RMSE errors, the `SZ3` compression algorithm should be chosen over





**Figure 13.** Comparison between psit and `ZFP` for the `tra_20000101_00` and `tra_20000101_00_permuted` files. The configuration of our program is described in Subsection 3.2 and the corresponding config file is given in Listing 1. The L1, RMSE and L-infinity error is compared to the achieved compression ratio for the central angle error (lon-lat) and the other data variables (pressure (p), temperature (T), potential vorticity (PV), specific humidity (Q)).





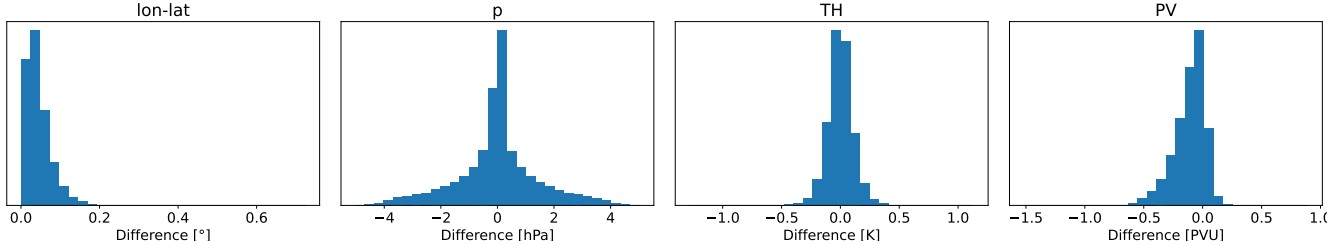

**Figure 14.** Error distribution based on values for the `tra_20200101_00` dataset using the compression setup of Subsection 3.2 with compression factor of 30. The data variables are central angle (lon-lat), pressure (p), potential temperature (TH), and potential vorticity (PV).

`JPEG 2000`. When looking at `ZFP`, we see that the performance between the different input data files differs significantly. The performance of the permuted trajectory file is much worse compared to the non permuted one. This is because, for the `tra_20200101_00` file, the trajectories have been initialized on an equidistant grid and are stored in the file in an ordered manner. Therefore, by converting them into the 2D grids over time steps and trajectories, a naive smoothness (originating from spatial continuity) of the grid is created. When we permute the trajectories, this naive smoothness is destroyed, and the compression performance drops. This means that the compression performance of `ZFP` does not only depend on the data, but also on the way in which it is stored, something from which psit does not suffer. This also makes it difficult to predict and quantify the behavior of `ZFP` on input files.

In the second run, we consider the `tra_20000101_00` and `tra_20000101_00_permuted` files. This can be seen in Fig. 13, with exact values given in Tables B4, B5, and B6. We observe a very similar, though less pronounced, behavior. Now, the difference between the two compression methods is less distinct, and psit only starts to perform better than `ZFP` for larger compression ratios. This decrease in performance for this larger trajectory file indicates that psit, in general, struggles with longer time ranges. A reason for this could be that over time, trajectories start to diverge, and therefore, the smoothness in the images, which is based on their initial position, gets lost.

From the above observations, we can make some general statements about the performance of psit. For smaller time ranges, psit performs on par with or better than `ZFP`, with a notable exception being the pressure variable, due to its inherent noisy behavior. For longer time ranges, the performance of psit starts to degrade. Additionally, for cases where keeping a small L-infinity error is important, we should use `SZ3` over `JPEG 2000`, as it bounds the L-infinity error. Moreover, the performance of `ZFP` is highly dependent on the way the input data is structured, something from which psit does not suffer. We therefore conclude that psit is a viable alternative to `SZ3` and can deliver consistent results.

### 3.2.2 Error Value Distribution

The next experiment we are conducting is about the error distribution. For this, we compress the `tra_20200101_00` and `tra_20000101_00` files with a compression factor of 30, then calculate the difference from the original file and put these





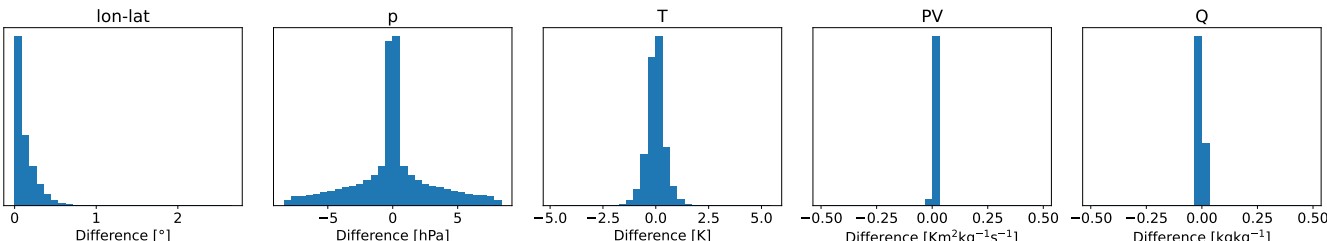

**Figure 15.** Error distribution based on values for the `tra_20000101_00` dataset using the compression setup of Subsection 3.2 with compression factor of 30. The data variables are central angle (lon-lat), pressure (p), temperature (T), potential vorticity (PV), and specific humidity (Q).

differences into a histogram. We would then expect the error distribution to follow a Gaussian, and depending on the actual shape of the histograms, we can gauge if bias is produced by the compression.

The histogram for the `tra_20200101_00` file is shown in Fig. 14. For the pressure and potential temperature variables, we

can observe a normal distribution of error values. In contrast, the potential vorticity variable is slightly skewed toward negative values, indicating that the compressed data tends to be larger than the original data. This is due to the large negative outliers present in the dataset. The Gaussian behavior of the pressure and potential temperature variables suggests that the compression procedure does not introduce any artificial bias in them, which could lead to systematic errors in subsequent research. For the `tra_20000101_00` file, shown in Fig. 15, we can observe an even distribution around 0 for all data variables. These

distributions are not exactly normally distributed, but we can still see them strictly decaying for larger error values.

## 3.3    Trajectory Density and Comparison to Perturbed Wind Fields

In this experiment, we examine the impact compression has on the position of the trajectories over time. To examine this impact, we calculate the densities of the trajectories starting at different pressure levels ($1\,\mathrm{hPa}$, $500\,\mathrm{hPa}$, $1000\,\mathrm{hPa}$) for increasing time steps ($72\,\mathrm{h}$, $120\,\mathrm{h}$, $168\,\mathrm{h}$). In addition to comparing uncompressed and compressed trajectories, we also look at trajectories

calculated from perturbed wind fields, simulating the impact of measurement uncertainties, and trajectories calculated from bitrounded wind fields. The bitrounding of the wind field is based on the paper by Klöwer et al. (2021) and was done using the `cdo bitrounding` operator (Schulzweida, 2023) and preserves $99.99\%$ of the mantissa information. In total, we consider the following trajectory datasets:

- Uncompressed trajectories,

- the `tra_20000101_00` trajectory file compressed with psit for compression ratios of $1.67$, $12.8$, $44.5$,

- trajectories (similar to `tra_20000101_00`) calculated from wind fields to which uniform noise to the horizontal/vertical components of magnitudes $0.01\,\mathrm{m\,s^{-1}}$ / $6.67\cdot10^{-4}\,\mathrm{Pa\,s^{-1}}$, $0.05\,\mathrm{m\,s^{-1}}$ / $3.33\cdot10^{-3}\,\mathrm{Pa\,s^{-1}}$, and $0.1\,\mathrm{m\,s^{-1}}$ / $6.67\cdot10^{-3}\,\mathrm{Pa\,s^{-1}}$ has been added,



## Trajectories starting at 1hPa

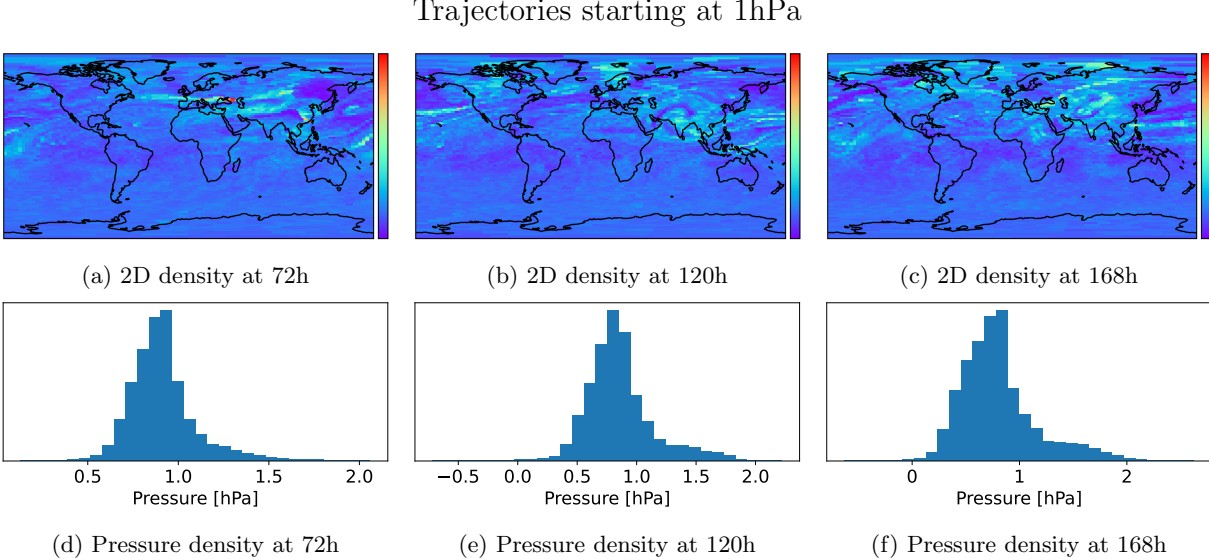

**Figure 16.** Density of uncompressed trajectories starting at $1\,\mathrm{hPa}$ over multiple time steps, the densities are given as a 2D density over longitude and latitude, and as a 1D density over pressure.

## Trajectories starting at 500hPa

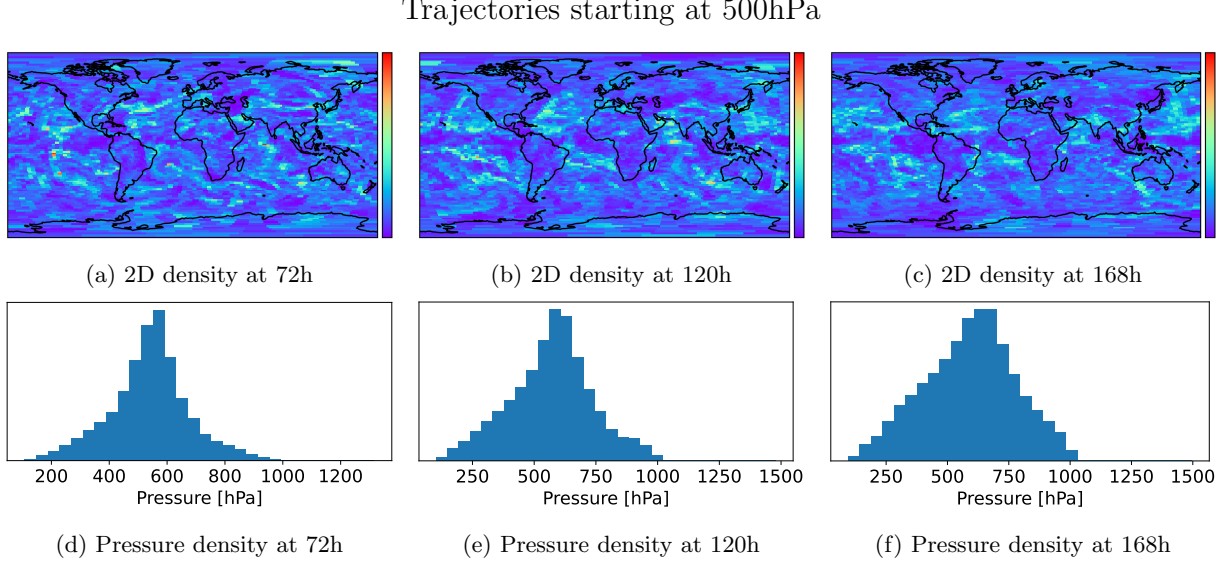

**Figure 17.** Density of uncompressed trajectories starting at $500\,\mathrm{hPa}$ over multiple time steps, the densities are given as a 2D density over longitude and latitude, and as a 1D density over pressure.



Trajectories starting at 1000hPa

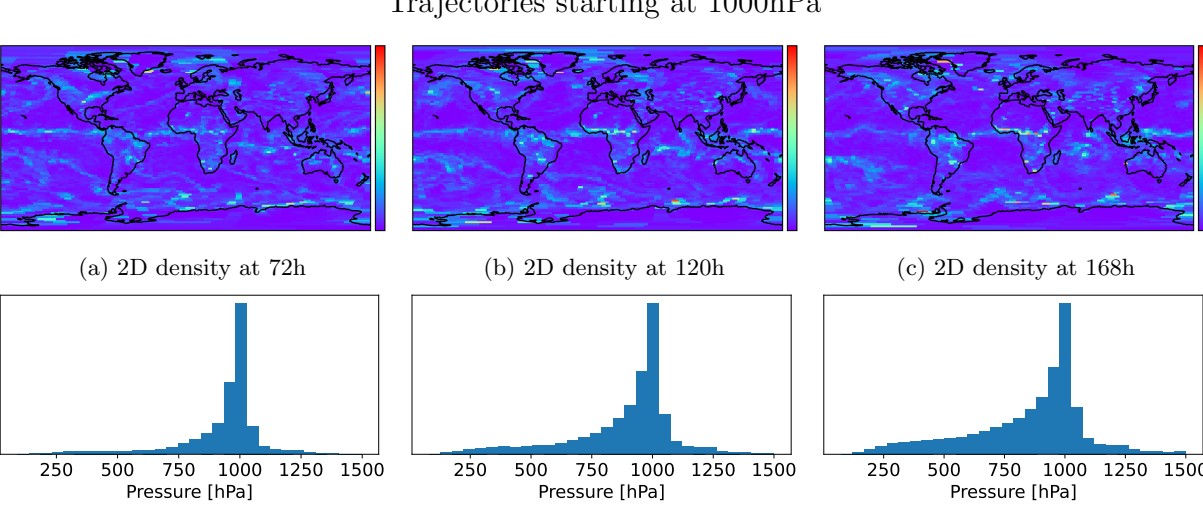

**Figure 18.** Density of uncompressed trajectories starting at $1000\,\mathrm{hPa}$ over multiple time steps, the densities are given as a 2D density over longitude and latitude, and as a 1D density over pressure.

- and trajectories (similar to `tra_20000101_00`) originating from bitrounded wind fields, with $99.99\%$ of the mantissa information retained.

We then calculate 2D densities over longitude and latitude (number trajectories over given area), and 1D densities over pressure (number trajectories over given pressure range), for which we consider the Jensen-Shannon divergence. We calculate the Jensen-Shannon divergence for the 2D case by partitioning the globe into equally sized bins and counting the number of trajectories per bin, and for the 1D case by using a histogram. Such a density comparison between these different datasets and the uncompressed one allows us to see the impact compression has on the trajectory positions and also how compression behaves compared to perturbed wind fields.

The density plots are given in Fig. 16, Fig. 17, and Fig. 18, with the Jensen-Shannon divergence shown in Tables B7, B8, and B9. From the tables, we can see that over time, the errors start to increase. The same (unsurprisingly) also holds for increasing compression ratio and perturbation magnitude. Furthermore, the error in pressure for the $500\,\mathrm{hPa}$ and $1000\,\mathrm{hPa}$ pressure levels tends to be large and very similar between all runs, indicating that it is a very sensitive metric, as any perturbation leads to large errors. For the 2D divergence metric, we find that with psit compressed trajectories in the compression range of $12.8$ to $44.5$ behave very similarly to perturbed wind fields with perturbation magnitudes of $0.05\,\mathrm{m\,s^{-1}}$ / $3.33\cdot10^{-3}\,\mathrm{Pa\,s^{-1}}$ to $0.1\,\mathrm{m\,s^{-1}}$ / $6.67\cdot10^{-3}\,\mathrm{Pa\,s^{-1}}$. The lowest compression ratio of $1.67$ tends to outperform all the perturbed wind fields. Additionally, it can be seen that the bitrounded wind fields behave very poorly, indicating that using bitrounding as a compression technique for wind fields could lead to problems. This experiment demonstrates that psit behaves very similarly to perturbed wind fields, showing that the error incurred by compression lies within inaccuracies generated by data assimilation techniques.



(a) Uncompressed data

(b) Psit 1.67x

(c) Psit 12.8x

(d) Psit 44.5x

(e) Difference psit 1.67x

(f) Difference psit 12.8x

(g) Difference psit 44.5x

(h) Perturbed $0.01\,\mathrm{ms}^{-1}$

(i) Perturbed $0.05\,\mathrm{ms}^{-1}$

(j) Perturbed $0.1\,\mathrm{ms}^{-1}$

(k) Difference perturbed $0.01\,\mathrm{ms}^{-1}$

(l) Difference perturbed $0.05\,\mathrm{ms}^{-1}$

(m) Difference perturbed $0.1\,\mathrm{ms}^{-1}$

(n) Bitrounding

**Figure 19.** Density of trajectories starting in the box from $0°\,\mathrm{E}$ to $15°\,\mathrm{E}$ times $45°\,\mathrm{N}$ to $60°\,\mathrm{N}$, next to the densities the difference to the uncompressed trajectories are plotted. Note that for the bitrounded example no difference plot is given because of its bad performance.





Instead of looking at global trajectory densities, we can also look at local ones. For this, we take all the trajectories starting in the box from $0°\,E$ to $15°\,E$ and $45°\,N$ to $60°\,N$, and plot their density alongside the difference to the uncompressed trajectories after $168\,h$ (Fig. 19). From these plots, we can see that the general features are preserved under compression and that

compression behaves very similarly to adding random perturbations to the wind field, again confirming that psit and perturbed wind fields lead to similar behavior of the trajectories.

### 3.4  Case Studies

#### 3.4.1  Warm Conveyor Belts

The Warm Conveyor Belt (WCB) is one of three characteristic air streams in extratropical cyclones (Browning et al., 1973;

Madonna et al., 2014). WCBs ascend in the cyclones' warm sector from the boundary layer to upper-tropospheric levels. This ascent, which occurs within 48 hours and extends over 600 hPa, is associated with substantial cloud and precipitation formation (Pfahl et al., 2014). The WCB also influences the large-scale circulation and the weather predictability downstream of their interaction with the upper-level flow (Grams et al., 2011; Rodwell et al., 2018).

In order to carry out this experiment, we use the `tra_20000101_00_permuted` file and select all the trajectories which,

in the first $48\,h$, rise by at least $600\,hPa$. We do this for uncompressed data, with psit compressed data (ratios of 1.67, 12.8, and 44.5), and with `ZFP` compressed data (ratios of 3.08, 11.52, and 36.91). This is displayed in Fig. 20. For psit, we have that for all compression ratios, the general location and shape of the WCBs remain the same. With increasing compression ratio, the number of false positives increases faster than the number of false negatives, with these false positives mostly located in regions with already existing WCB activity. This asymmetric behavior is most likely due to the fact that the dense data

compression algorithms perform smoothing of the data. Specifically, if a region with the majority of the selected trajectories gets smoothed out, the few unselected trajectories will start to behave like the selected ones, i.e., they will also be selected, leading to an increase in false positives. For `ZFP`, we can see a similar behavior of more false positives compared to false negatives in the medium compressed case. On the other hand, compared to psit, the number of misclassified trajectories is smaller, but the horizontal location of the trajectories is less accurate. Hence, we can see the WCB regions being smoothed

out, and for the most compressed case, very strong compression artifacts start to appear. This observation is in line with the previous one that psit struggles with pressure while delivering good results for longitude and latitude. But still, in all cases, we see that psit preserves the general structure, and only a very limited amount of false WCB activity gets generated due to compression (e.g., the region between the Southeast Atlantic Ocean and the Davis Sea).

#### 3.4.2  The Fukushima accident

During the Fukushima Daiichi accident (int, 2015) in 2011, radiation-contaminated material was released into the atmosphere. This leads to the question of where this radioactive material is transported and deposited (Wotawa, 2011; Chino et al., 2011; Katata et al., 2012; Yasunari et al., 2011; Stohl et al., 2012). In this experiment, we want to discuss the impact compression



(a) Uncompressed data

(b) Psit 1.67x

(c) Psit 12.8x

(d) Psit 44.5x

FP: 6, FN: 8

FP: 712, FN: 131

FP: 4957, FN: 220

(e) Difference psit 1.67x

(f) Difference psit 12.8x

(g) Difference psit 44.5x

(h) ZFP 3.08x

(i) ZFP 11.52x

(j) ZFP 36.91x

FP: 0, FN: 0

FP: 200, FN: 84

FP: 1586, FN: 3354

(k) Difference ZFP 3.08x

(l) Difference ZFP 11.52x

(m) Difference ZFP 36.91x

**Figure 20.** Warm conveyor belt calculations on the uncompressed data and the data compressed with different compression ratios. Additionally the difference in term of false positive and false negatives is shown.



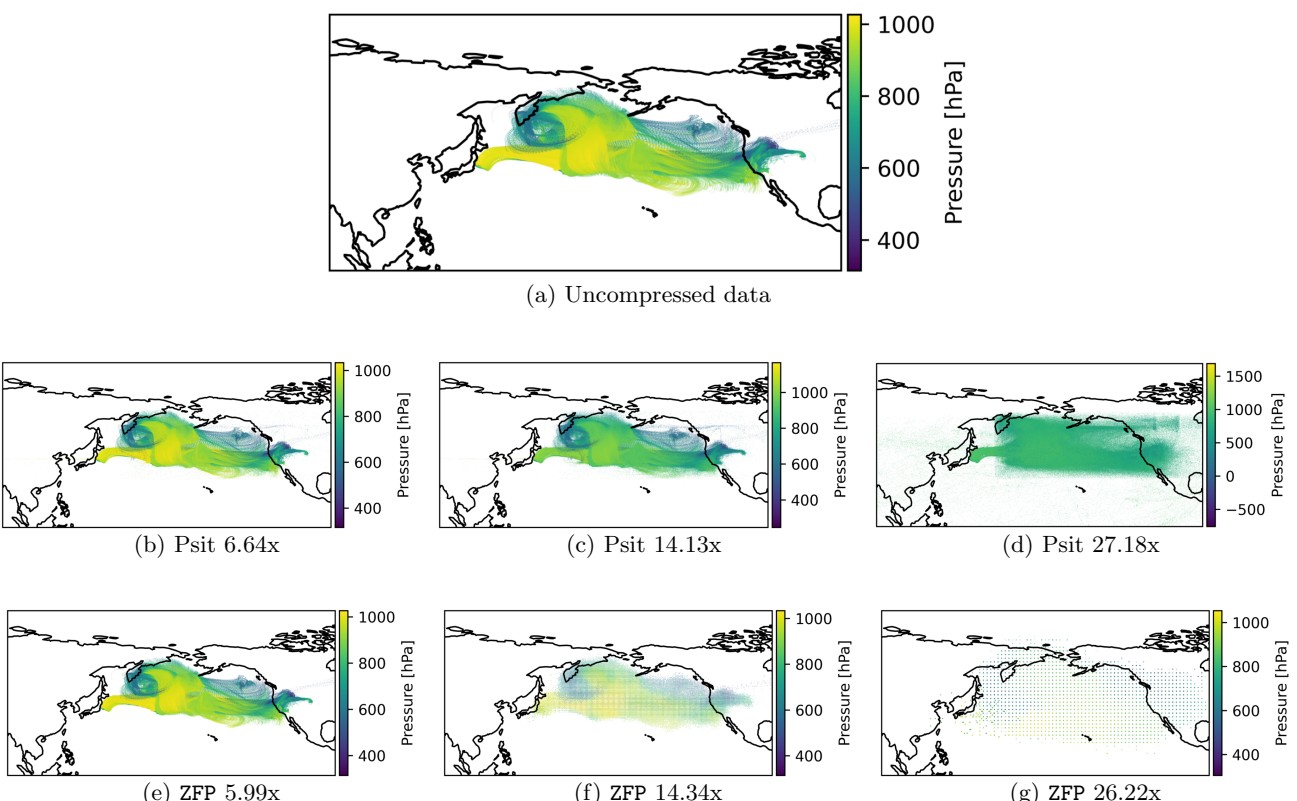

**Figure 21.** Comparison between the trajectories of the nuclear disaster at Fukushima for different compression ratios. On the top is the uncompressed data with, below it we have two rows of compressed data, one for psit and one for `ZFP`.

has on trajectories originating in the region of the Fukushima Daiichi power plant. We will not perform any detailed deposition analysis.

For this, we create $35'280$ trajectories starting at 2011-03-12 00:00 in the box $140.74°$ E to $141.3°$ E and $37.19°$ N to $37.63°$ N over the pressure range of $764\,\text{hPa}$ to $996\,\text{hPa}$. The only information stored on the trajectories is their position, leading to a $59\,\text{MB}$ file. For compression with psit, we choose `JPEG 2000` with XYZ color and delta encoding. Note that here we compress trajectories that originate in a local region instead of a global one. We then compress the trajectories using psit with three different compression ratios (6.64, 14.13, 27.18) and `ZFP` with three different compression ratios (5.99, 14.13, 26.22). These compression ratios have been chosen to showcase both cases where compression works and cases where compression breaks down. We then plot the trajectories over time, as shown in Fig. 21. From the panels, we can see that for both psit and `ZFP`, with increasing compression ratio, the results appear to get smoothed out and compression artifacts start to appear. We can also see that for psit the compression appears to perform worse, especially compared to Section 3.3. This is because the resulting grids have much smaller dimensions ($48 \times 37$ grid points), leading to less data per grid, which results in poorer





performance of the dense data compression algorithms. If we compare psit to `ZFP` we can see that for lower compression ratios they are very similar to one another and only at the larger compression ratios can we observe psit delivering better results. This is in line with previous observations. This demonstrates that while psit is able to preserve the general structure and regions that would be affected, it is best used in the high ratio compression of files where many trajectories are present.

### 3.5   Throughput and Memory Usage

Until now, we have evaluated the performance of the program from an error standpoint. Here, we focus on performance from a computational perspective, discussing both throughput and memory usage. For this, the `tra_20000101_00` trajectory file is compressed in the same manner as presented in Section 3.2. This is performed on an `AMD EPYC 7742` parallelized over 37 threads (for both compression and decompression), using a ramdisk to filter out the impact of a slow filesystem. We then calculate the throughput in MB/s for compression and decompression across the different compression ratios.

The results of the throuput experiments are presented in Table 1. The mean throughput of the compression stage is $65.67\,\mathrm{MB/s}$, while the mean throughput of decompression is $222.14\,\mathrm{MB/s}$. Decompression has a significantly larger throughput compared to compression because the expensive mapping stage only needs to be performed during compression. Another observation is that throughput increases as the compression factor gets larger. This is because a larger compression factor results in smaller file sizes, which, in turn, reduces the time needed to read and write data to disk. The memory usage for compression

with factor 30 was $60\,\mathrm{GB}$. The part where most of the memory consumption comes from is in solving the minimal weight full bipartite matching, which is parallelized over multiple workers. Therefore, one can trade memory usage against runtime by decreasing the number of parallel workers.




**Table 1.** Throughput for compression and decompression when running psit parallelized over 37 workers using a ramdisk.

| Factor | Compression | Decompression |
|--------|-------------|---------------|
| 1 | 32.47 MB/s | 144.67 MB/s |
| 5 | 39.25 MB/s | 159.53 MB/s |
| 10 | 57.04 MB/s | 205.73 MB/s |
| 15 | 61.65 MB/s | 221.90 MB/s |
| 20 | 66.82 MB/s | 232.07 MB/s |
| 25 | 69.64 MB/s | 238.83 MB/s |
| 30 | 71.82 MB/s | 239.78 MB/s |
| 40 | 72.27 MB/s | 246.00 MB/s |
| 50 | 76.36 MB/s | 242.35 MB/s |
| 75 | 78.77 MB/s | 246.95 MB/s |
| 100 | 80.48 MB/s | 249.27 MB/s |
| 150 | 81.47 MB/s | 238.55 MB/s |

## 4 Discussion

### 4.1 Applicable range and limitations

The main structural limitation of the program is the fact that the input data trajectories need to be uniformly distributed. This limitation originates from the way the mapping procedure has been implemented and has been touched upon in Section 2.5. The optimal fix for this limitation would be if the minimal weight full bipartite matching problem could be solved on a global level without having to restrict the grid points a trajectory may map to its closest 200 neighbors. A workaround for this is to artificially increase the ratio of grid points to trajectories, in this way, a mapping can be created at the cost of increased grid size, and because much of the data in the grids is redundant, the compression algorithms might be able to compress these grids more effectively, leading to only a small increase in output size.

In addition to the structural limitations, there are also performance limitations, especially memory consumption and runtime. Here, the bottleneck is the solving of the minimal weight full bipartite matching problem in the mapping phase. The only way to solve this would be to either use a different mapping method or find a more efficient implementation of such an algorithm. Additionally, all the images are currently saved individually, leading to the creation of potentially thousands of files, which can be slow on certain filesystems. Furthermore, currently the compression pipeline is written in Python and was developed parallel to the exploration of methods, leading to sometimes non optimal implementations.

Based on the above discussed limitations of the implementation, we argue that the applicable range of psit is in the compression of uniformly distributed trajectories in an HPC environment.





## 4.2 Future Work

Of course, one obvious area of future work would be fixing the current limitations of psit. This would include removing the limitation of a trajectory only being able to map to a set of its closest neighboring grid points and generally improving the overall computational performance of the pipeline in terms of runtime and memory usage. Fixing these limitations would be
quite beneficial for psit, as it would allow it to be used in a wider range of applications.

The mapping method currently used by psit is based on graph theory, specifically the solution to a bipartite matching problem. However, there are likely other methods by which this could be done. For instance, instead of solving the problem directly in one step, an initial simple mapping (such as via nearest neighbors) could be created, which is then iteratively refined to fulfill the initial constraints, with the goal being to create a valid mapping that is also smooth. Further exploration of what
"smooth" means could also be valuable. Currently, smoothness is achieved by using the trajectory positions at the first time step. What would happen if the mean position were used instead? Alternatively, could the smoothness of a grid be quantized to create a mapping that aims to optimize it? As the mapping phase is integral to the entire pipeline and currently presents the performance bottleneck, exploring these questions would be greatly beneficial.

The methods we use to describe compression performance are error metrics, comparison with perturbed/bitrounded wind
fields, and real world examples in the form of two case studies. However, there are some intricacies to consider. Error metrics provide a straightforward way to quantify the incurred error, but they often lack a strong connection to real-world applications. It can be challenging to predict the impact of compression on real-world research based solely on a simple error metric. Is an RMSE error of $0.5$ hPa in pressure acceptable, or should it be less than $0.3$ hPa? Moreover, simple error metrics do not illustrate the relationship between different variables. How does the error in the longitude variable compare to the error in the latitude
variable? Are they similar, or is there more error in one than the other? Does the error in the temperature variable depend on the pressure variable? A statistical analysis could examine correlations between various data variables, but this can quickly become tedious while leaving the fundamental question unanswered of how much correlation is acceptable. On the other hand, case studies simulate scenarios where the data might be used. We believe this approach establishes a better foundation for understanding the impact of compression. However, the challenge is that one can only analyze singular examples, and the
generalizability of findings from one test to another remains an open question. While writing this paper, we spent considerable time discussing with experts in the field how such errors could be described and interpreted, there we found three major points that we think are important: compression should not change the outcome of experiments, compression errors should be smaller than measurement errors, compression should not influence the timings and occurrence of physical processes (e.g., cloud formation). But even these three points are not universally applicable, e.g., how does one compare measurement errors
to compression errors, which physical processes are important enough to be conserved and to which degree should they be conserved. We tried to address these challenges in this paper, but we think that it would be beneficial and interesting to dedicate time into researching the possibilities of developing a benchmark that can help in understanding the impact of these errors.



This benchmark could combine error metrics with a list of real-world applications, providing users with a general overview of how compression performs across different aspects.

## 5   Conclusions

The initial problem we faced and aimed to solve was that increasing computational power leads to an increase in data, making storage unfeasible. Therefore, a method needs to be devised to reduce it, with one approach being compression. While compression schemes exist for the Eulerian frame, no equivalent alternatives exist for the Lagrangian one. We therefore developed psit, a system to compress Lagrangian flows, to address this issue. In this paper, we presented how psit works and evaluated its performance using error metrics and case studies. In these, we demonstrated that in most cases, compression performance equivalent to or superior to ZFP can be achieved. We showed that the trajectory density of compressed trajectories (ratio 30 to 40) after 168 hours behaves similarly to uncompressed trajectories calculated from perturbed wind fields (with a perturbation magnitude of $0.1\,\mathrm{m\,s^{-1}}$ / $6.67\cdot10^{-3}\,\mathrm{Pa\,s^{-1}}$). Additionally, we analyzed the impact of compression on subsequent research through two case studies: warm conveyor belts and fallout prediction of the Fukushima accident. While psit imposes some limitations on the input data, namely requiring a uniform initial distribution, it can be used for general purpose compression of Lagrangian weather data and can serve as a foundation for further research in this area.

*Code availability.*   The version of psit referenced in this paper is available at https://doi.org/10.5281/zenodo.14888490 (Pietak, 2025a). An up do date version can be found on github https://github.com/apietak/psit (Pietak, 2025b).

## Appendix A:  Proof of Total Unimodularity

In this section, we provide a proof that the LP presented in the LP mapping of Subsection 2.2 is an integral LP. An LP of the form $\{\min \boldsymbol{c}^T \boldsymbol{x} \mid \mathbf{A}\boldsymbol{x} \geq \boldsymbol{b},\ \boldsymbol{x} \geq \boldsymbol{0}\}$ is integral if the constraint matrix $\mathbf{A}$ is totally unimodular and the right hand side vector $\boldsymbol{b}$ is integral. In this mapping, the $\boldsymbol{b}$ vector consists solely of values equal to 1, making it integral. The only remaining step is to show that the matrix $\mathbf{A}$ is totally unimodular.

Before we begin the proof, we must discuss the structure of the matrix $\mathbf{A}$. In the definition of the LP, it was stated that the vector $\boldsymbol{x}$ consists of the values $x_{t,p}$ with $t \in T$ and $p \in P$, and that there are two types of constraints:

$$\sum_{t \in N^-(p)} x_{t,p} = 1 \quad \forall p \in P \tag{A1}$$

$$\sum_{p \in N^+(t)} x_{t,p} \geq 1 \quad \forall t \in T \tag{A2}$$

Note that in contrast to the definition given in Subsection 2.2, here we also have the terms $N^-(p)$ and $N^+(t)$ which denote the possible matchings of trajectories to grid points, i.e., $N^-(p)$ for all the trajectories which may map to a point $p$ and $N^+(t)$





for all the grid points which can be mapped by a trajectory $t$. This is such that the proof is also valid for the shortcuts we took during implementation. To transform these constraints into canonical form, we need to rewrite the equality constraint as a set of two inequalities. The relationship $a = b$ can be expressed as $a \geq b$ and $-a \geq -b$. Thus, the constraints become:

$$\sum_{t \in N^-(p)} x_{t,p} \geq 1 \quad \forall p \in P \tag{A3}$$

$$\sum_{t \in N^-(p)} -x_{t,p} \geq -1 \quad \forall p \in P \tag{A4}$$

$$\sum_{p \in N^+(t)} x_{t,p} \geq 1 \quad \forall t \in T \tag{A5}$$

Next, we need to express these constraints in matrix form. To do this, we must establish a representation for the vector $\boldsymbol{x}$. Without loss of generality, we can assume that $\boldsymbol{x}$ is constructed by first looping over all trajectories and then looping over all grid points to which each trajectory may map. This means that blocks of $x_{t,p}$ with the same $t$ and varying $p$ are stacked on top of each other. For example, if we have three trajectories $t_1, t_2, t_3$ and four grid points $p_1, p_2, p_3, p_4$, the resulting vector $\boldsymbol{x}$ could look like this:

$$\boldsymbol{x} = \begin{bmatrix} x_{t_1,p_2} \\ x_{t_1,p_3} \\ x_{t_2,p_1} \\ x_{t_2,p_3} \\ x_{t_3,p_4} \end{bmatrix} \tag{A6}$$

With the shape of the vector $\boldsymbol{x}$ established, we can now construct the matrix $\mathbf{A}$. This matrix consists of two parts. In the first part, called $\mathbf{A}_1$, we encode the first two constraints (Eqs. (A3) and (A4)), and in the second part, called $\mathbf{A}_2$, we encode constraint Eq. (A5).

For the matrix $\mathbf{A}_1$, each row corresponds to a grid point. A row resulting from constraint Eq. (A3) consists of 1s and 0s, with at most one 1 per block of the same trajectory in $\boldsymbol{x}$. The exact distribution of these 1s depends on the shape of $\boldsymbol{x}$, but what is important is that no two rows have a 1 in the same column. This follows directly from the structure of $\boldsymbol{x}$ and the fact that each row corresponds to a grid point. Another important property is that when constraints Eq. (A3) and (A4) are combined, there are always pairs of rows that are identical, except that the sign of one row is the opposite of the other. Using the same example





as above, the matrix $\mathbf{A}_1$ will look like this:

$$
\mathbf{A}_1 =
\begin{bmatrix}
0 & 0 & 1 & 0 & 0 \\
0 & 0 & -1 & 0 & 0 \\
1 & 0 & 0 & 0 & 0 \\
-1 & 0 & 0 & 0 & 0 \\
0 & 1 & 0 & 1 & 0 \\
0 & -1 & 0 & -1 & 0 \\
0 & 0 & 0 & 0 & 1 \\
0 & 0 & 0 & 0 & -1
\end{bmatrix}
\tag{A7}
$$

Here, the pairs of rows with opposite signs are clearly visible.

Matrix $\mathbf{A}_2$ results from constraint Eq. (A5). In this case, each row corresponds to a trajectory, and all columns are set to 1 where a corresponding $x_{t,p}$ variable exists for that specific trajectory. Due to the construction of the $\boldsymbol{x}$ vector, all the different trajectories are grouped into blocks, and for each block, all the grid points that the trajectory can map to are indicated. Consequently, each row will contain a continuous sequence of 1s corresponding to the $x_{t,p}$ values of the $\boldsymbol{x}$ vector with a specific $t$. This matrix will have a staircase structure, ensuring that no two rows have a 1 in the same column. Combining the previous part of $\mathbf{A}$ with this new one, it can be defined as follows:

$$
\mathbf{A} =
\begin{bmatrix}
\mathbf{A}_1 \\
\mathbf{A}_2
\end{bmatrix}
=
\begin{bmatrix}
0 & 0 & 1 & 0 & 0 \\
0 & 0 & -1 & 0 & 0 \\
1 & 0 & 0 & 0 & 0 \\
-1 & 0 & 0 & 0 & 0 \\
0 & 1 & 0 & 1 & 0 \\
0 & -1 & 0 & -1 & 0 \\
0 & 0 & 0 & 0 & 1 \\
0 & 0 & 0 & 0 & -1 \\
1 & 1 & 0 & 0 & 0 \\
0 & 0 & 1 & 1 & 0 \\
0 & 0 & 0 & 0 & 1
\end{bmatrix}
\tag{A8}
$$

The last step is to prove that this matrix is totally unimodular. The definition of total unimodularity states that for every square submatrix $\mathbf{S}$ of $\mathbf{A}$, its determinant must be either $-1$, $0$, or $1$. Proving this directly can be quite challenging. However, Ghouila-Houri has shown that a matrix is totally unimodular if and only if for every subset $R$ of rows of the matrix, there exists an assignment $s : R \rightarrow \pm$ of signs such that, when summed, every element in the resulting vector is either $-1$, $0$, or $1$. Therefore, we must find such a mapping $s : R \rightarrow \pm$ to conclude the proof.





To define this mapping, we use the property that in both parts of the matrix, no two rows have a $1$ with the same sign in the same column. Initially, we assume that constraint Eq. (A4) does not exist, leading to the following definition of the mapping:

$$\forall r \in R: \quad s(r) = \begin{cases} + & \text{if } r \in \mathbf{A}_1 \\ - & \text{if } r \in \mathbf{A}_2 \end{cases} \tag{A9}$$

We will first sum the vectors separately for the two parts, $\mathbf{A}_1$ and $\mathbf{A}_2$. Due to the previously mentioned property, the vector derived from $\mathbf{A}_1$ will contain only elements from $\{0, 1\}$, while the vector from $\mathbf{A}_2$ will consist only of elements from $\{-1, 0\}$.

The sum of these two vectors will thus produce a vector with elements from $\{-1, 0, 1\}$, confirming that a valid mapping has been found.

Next, we will reintroduce constraint Eq. (A4) into the formulation. The mapping can be adjusted to accommodate this case by making a small extension: a row $r \in R$ resulting from Eq. (A4) will be mapped to a $-$ if and only if the same row with the opposite sign is *not* in $R$. In this way, two scenarios can occur: either both the positive and negative variants of a row are in $R$,

in which case they will cancel out, or only one of them, either the negative or the positive variant, is in $R$. In the latter case, a $-$ would be assigned to the negative variant, transforming the problem into the same form as if constraint Eq. (A4) did not exist at all. Therefore, it has been demonstrated that for any arbitrary subset $R$ of rows from $\mathbf{A}$, a mapping $s : R \to \pm$ has been found that makes the matrix totally unimodular, thereby proving that the LP is integral.    □

**Appendix B: Performance tables of psit and `ZFP`**





**Table B1.** Error values and compression ratios for psit on the `tra_20200101_00` dataset.

| Factor | Ratio | lon-lat | | | p | | | TH | | | PV | | |
|---|---|---|---|---|---|---|---|---|---|---|---|---|---|
| | | L1 | RMSE | L-infinity | L1 | RMSE | L-infinity | L1 | RMSE | L-infinity | L1 | RMSE | L-infinity |
| 1 | 2.08 | 0.10* | 0.17* | 0.65 | 0.02 | 0.03 | 0.11 | 0.22* | 0.43* | 0.07 | 0.12 | 0.17 | 0.71 |
| 5 | 2.55 | 0.11* | 0.18* | 0.72 | 0.17 | 0.24 | 0.78 | 0.24* | 0.43* | 0.07 | 0.12 | 0.17 | 0.71 |
| 10 | 5.04 | 0.65* | 0.84* | 0.72 | 0.35 | 0.51 | 1.68 | 0.02 | 0.02 | 0.25 | 0.12 | 0.17 | 0.78 |
| 15 | 6.58 | 0.01 | 0.02 | 0.74 | 0.51 | 0.74 | 2.46 | 0.03 | 0.04 | 0.48 | 0.12 | 0.17 | 0.92 |
| 20 | 8.63 | 0.02 | 0.03 | 0.73 | 0.69 | 0.99 | 3.35 | 0.05 | 0.07 | 0.83 | 0.13 | 0.17 | 1.14 |
| 25 | 9.91 | 0.03 | 0.04 | 0.75 | 0.83 | 1.21 | 4.13 | 0.06 | 0.09 | 1.05 | 0.13 | 0.17 | 1.26 |
| 30 | 11.61 | 0.04 | 0.05 | 0.72 | 0.99 | 1.45 | 5.03 | 0.08 | 0.11 | 1.28 | 0.13 | 0.17 | 1.52 |
| 40 | 14.20 | 0.06 | 0.08 | 0.88 | 1.26 | 1.87 | 6.70 | 0.11 | 0.14 | 1.94 | 0.13 | 0.18 | 1.71 |
| 50 | 16.49 | 0.08 | 0.10 | 1.10 | 1.51 | 2.26 | 8.38 | 0.13 | 0.17 | 2.32 | 0.13 | 0.18 | 2.07 |
| 75 | 21.11 | 0.11 | 0.14 | 1.70 | 2.06 | 3.14 | 12.51 | 0.16 | 0.22 | 3.10 | 0.14 | 0.19 | 3.29 |
| 100 | 25.02 | 0.14 | 0.18 | 2.50 | 2.55 | 3.93 | 16.75 | 0.19 | 0.27 | 4.55 | 0.14 | 0.20 | 3.38 |
| 150 | 30.63 | 0.18 | 0.23 | 3.60 | 3.36 | 5.30 | 25.13 | 0.23 | 0.33 | 5.96 | 0.15 | 0.22 | 5.30 |

* Values scaled by 100

**Table B2.** Error values and compression ratios of the `ZFP` baseline run on the `tra_20200101_00` dataset. Note that for some different tolerances the compression ratios are the same, this appears to be related to how `ZFP` works.

| Tolerance | Ratio | lon-lat | | | p | | | TH | | | PV | | |
|---|---|---|---|---|---|---|---|---|---|---|---|---|---|
| | | L1 | RMSE | L-infinity | L1 | RMSE | L-infinity | L1 | RMSE | L-infinity | L1 | RMSE | L-infinity |
| 0.01 | 2.90 | 0.07* | 0.08* | 0.40* | 0.05* | 0.07* | 0.36* | 0.05* | 0.07* | 0.35* | 0.05* | 0.07* | 0.36* |
| 0.1 | 4.10 | 0.55* | 0.64* | 0.03 | 0.41* | 0.52* | 0.03 | 0.41* | 0.52* | 0.03 | 0.40* | 0.51* | 0.03 |
| 0.5 | 6.64 | 0.04 | 0.04 | 0.25 | 0.03 | 0.04 | 0.23 | 0.03 | 0.04 | 0.22 | 0.03 | 0.03 | 0.37 |
| 0.05 | 3.61 | 0.28* | 0.33* | 0.02 | 0.20* | 0.26* | 0.01 | 0.20* | 0.26* | 0.01 | 0.20* | 0.26* | 0.01 |
| 1.0 | 8.11 | 0.07 | 0.08 | 0.48 | 0.06 | 0.08 | 0.47 | 0.06 | 0.07 | 0.46 | 0.04 | 0.06 | 0.74 |
| 2.0 | 10.13 | 0.12 | 0.14 | 0.83 | 0.13 | 0.16 | 0.91 | 0.10 | 0.13 | 0.89 | 0.07 | 0.10 | 1.44 |
| 3.0 | 10.13 | 0.12 | 0.14 | 0.83 | 0.13 | 0.16 | 0.91 | 0.10 | 0.13 | 0.89 | 0.07 | 0.10 | 1.44 |
| 4.0 | 13.07 | 0.21 | 0.24 | 1.57 | 0.24 | 0.31 | 1.79 | 0.18 | 0.23 | 1.62 | 0.12 | 0.16 | 1.55 |
| 5.0 | 13.07 | 0.21 | 0.24 | 1.57 | 0.24 | 0.31 | 1.79 | 0.18 | 0.23 | 1.62 | 0.12 | 0.16 | 1.55 |
| 7.0 | 13.07 | 0.21 | 0.24 | 1.57 | 0.24 | 0.31 | 1.79 | 0.18 | 0.23 | 1.62 | 0.12 | 0.16 | 1.55 |
| 10.0 | 17.66 | 0.36 | 0.42 | 3.01 | 0.46 | 0.59 | 3.53 | 0.30 | 0.38 | 3.26 | 0.19 | 0.25 | 3.21 |
| 15.0 | 17.66 | 0.36 | 0.42 | 3.01 | 0.46 | 0.59 | 3.53 | 0.30 | 0.38 | 3.26 | 0.19 | 0.25 | 3.21 |
| 20.0 | 25.53 | 0.70 | 0.81 | 6.11 | 0.83 | 1.08 | 7.29 | 0.46 | 0.59 | 5.62 | 0.30 | 0.40 | 6.35 |

* Values scaled by 100





**Table B3.** Error values and compression ratios of the `ZFP` baseline run on the `tra_20200101_00_permuted` dataset. Note that for some different tolerances the compression ratios are the same, this appears to be related to how `ZFP` works.

| Tolerance | Ratio | lon-lat | | | p | | | TH | | | PV | | |
|---|---|---|---|---|---|---|---|---|---|---|---|---|---|
| | | L1 | RMSE | L-infinity | L1 | RMSE | L-infinity | L1 | RMSE | L-infinity | L1 | RMSE | L-infinity |
| 0.01 | 2.03 | 0.07 | 0.08 | 0.40 | 0.05 | 0.07 | 0.36 | 0.05 | 0.07 | 0.37 | 0.05 | 0.07 | 0.35 |
| 0.1 | 2.55 | 0.57 | 0.67 | 0.03 | 0.41 | 0.52 | 0.03 | 0.41 | 0.52 | 0.03 | 0.41 | 0.52 | 0.03 |
| 0.5 | 3.47 | 0.04 | 0.05 | 0.24 | 0.03 | 0.04 | 0.23 | 0.03 | 0.04 | 0.23 | 0.03 | 0.04 | 0.23 |
| 0.05 | 2.35 | 0.29 | 0.34 | 0.02 | 0.20 | 0.26 | 0.01 | 0.20 | 0.26 | 0.01 | 0.20 | 0.26 | 0.01 |
| 1.0 | 3.93 | 0.08 | 0.10 | 0.51 | 0.07 | 0.08 | 0.46 | 0.07 | 0.08 | 0.45 | 0.06 | 0.07 | 0.73 |
| 2.0 | 4.52 | 0.16 | 0.18 | 0.87 | 0.13 | 0.17 | 0.94 | 0.13 | 0.16 | 0.93 | 0.10 | 0.13 | 1.50 |
| 3.0 | 4.52 | 0.16 | 0.18 | 0.87 | 0.13 | 0.17 | 0.94 | 0.13 | 0.16 | 0.93 | 0.10 | 0.13 | 1.50 |
| 4.0 | 5.24 | 0.29 | 0.34 | 1.72 | 0.26 | 0.33 | 1.88 | 0.23 | 0.29 | 1.84 | 0.17 | 0.22 | 2.95 |
| 5.0 | 5.24 | 0.29 | 0.34 | 1.72 | 0.26 | 0.33 | 1.88 | 0.23 | 0.29 | 1.84 | 0.17 | 0.22 | 2.95 |
| 7.0 | 5.24 | 0.29 | 0.34 | 1.72 | 0.26 | 0.33 | 1.88 | 0.23 | 0.29 | 1.84 | 0.17 | 0.22 | 2.95 |
| 10.0 | 6.14 | 0.56 | 0.65 | 3.29 | 0.52 | 0.66 | 3.71 | 0.37 | 0.47 | 3.43 | 0.29 | 0.37 | 5.78 |
| 15.0 | 6.14 | 0.56 | 0.65 | 3.29 | 0.52 | 0.66 | 3.71 | 0.37 | 0.47 | 3.43 | 0.29 | 0.37 | 5.78 |
| 20.0 | 7.28 | 1.06 | 1.25 | 7.17 | 1.02 | 1.30 | 7.25 | 0.61 | 0.78 | 6.11 | 0.53 | 0.67 | 6.25 |

[*] Values scaled by 100

**Table B4.** Error values and compression ratios for psit on the `tra_20000101_00` dataset.

| Factor | Ratio | lon-lat | | | p | | | temperature | | | potential vorticity | | | specific humidity | | |
|---|---|---|---|---|---|---|---|---|---|---|---|---|---|---|---|---|
| | | L1 | RMSE | L-infinity | L1 | RMSE | L-infinity | L1 | RMSE | L-infinity | L1 | RMSE | L-infinity | L1 | RMSE | L-infinity |
| 1 | 1.67 | 0.05[*] | 0.28[*] | 2.50 | 0.05 | 0.07 | 0.19 | 0.63[*] | 0.04 | 0.92 | 0.01[*] | 0.31[*] | 0.49 | 0.01[*] | 0.31[*] | 0.49 |
| 5 | 2.36 | 0.09[*] | 0.29[*] | 2.53 | 0.32 | 0.48 | 1.30 | 0.71[*] | 0.04 | 0.92 | 0.01[*] | 0.31[*] | 0.49 | 0.01[*] | 0.31[*] | 0.49 |
| 10 | 5.25 | 0.02 | 0.02 | 2.61 | 0.69 | 1.03 | 2.78 | 0.05 | 0.08 | 1.04 | 0.01[*] | 0.31[*] | 0.49 | 0.01[*] | 0.31[*] | 0.49 |
| 15 | 6.98 | 0.03 | 0.05 | 2.49 | 1.01 | 1.51 | 4.08 | 0.11 | 0.15 | 2.03 | 0.01[*] | 0.31[*] | 0.49 | 0.01[*] | 0.31[*] | 0.49 |
| 20 | 9.29 | 0.07 | 0.10 | 2.57 | 1.38 | 2.06 | 5.56 | 0.19 | 0.25 | 3.02 | 0.01[*] | 0.31[*] | 0.49 | 0.01[*] | 0.31[*] | 0.49 |
| 25 | 10.77 | 0.09 | 0.13 | 2.46 | 1.69 | 2.53 | 6.86 | 0.24 | 0.32 | 3.98 | 0.01[*] | 0.31[*] | 0.49 | 0.01[*] | 0.31[*] | 0.49 |
| 30 | 12.79 | 0.13 | 0.19 | 2.66 | 2.05 | 3.08 | 8.35 | 0.31 | 0.42 | 5.44 | 0.01[*] | 0.30[*] | 0.49 | 0.01[*] | 0.30[*] | 0.49 |
| 40 | 15.98 | 0.19 | 0.27 | 3.52 | 2.72 | 4.08 | 11.13 | 0.41 | 0.58 | 7.42 | 0.01[*] | 0.30[*] | 0.49 | 0.02[*] | 0.30[*] | 0.50 |
| 50 | 18.98 | 0.25 | 0.36 | 4.08 | 3.36 | 5.07 | 13.91 | 0.51 | 0.72 | 9.12 | 0.01[*] | 0.32[*] | 0.76 | 0.02[*] | 0.32[*] | 0.68 |
| 75 | 25.71 | 0.39 | 0.55 | 7.57 | 4.90 | 7.46 | 20.77 | 0.70 | 1.02 | 12.48 | 0.01[*] | 0.31[*] | 0.68 | 0.02[*] | 0.31[*] | 0.72 |
| 100 | 32.37 | 0.53 | 0.76 | 11.31 | 6.41 | 9.83 | 27.82 | 0.88 | 1.31 | 18.55 | 0.01[*] | 0.34[*] | 1.23 | 0.02[*] | 0.34[*] | 1.23 |
| 150 | 44.55 | 0.78 | 1.14 | 16.94 | 9.20 | 14.32 | 41.73 | 1.15 | 1.75 | 27.27 | 0.02[*] | 0.01 | 4.01 | 0.04[*] | 0.01 | 4.01 |

[*] Values scaled by 100



**Table B5.** Error values and compression ratios of the `ZFP` baseline run on the `tra_20000101_00` dataset. Note that for some different tolerances the compression ratios are the same, this appears to be related to how `ZFP` works.

| Tolerance | Ratio | lon-lat | | | p | | | temperature | | | potential vorticity | | | specific humidity | | |
|---|---|---|---|---|---|---|---|---|---|---|---|---|---|---|---|---|
| | | L1 | RMSE | L-infinity | L1 | RMSE | L-infinity | L1 | RMSE | L-infinity | L1 | RMSE | L-infinity | L1 | RMSE | L-infinity |
| 0.1 | 5.49 | 0.07* | 0.09* | 0.43* | 0.05* | 0.07* | 0.37* | 0.05* | 0.07* | 0.37* | 0.00* | 0.01* | 0.55* | 0.01* | 0.02* | 0.58* |
| 0.5 | 8.19 | 0.55* | 0.64* | 0.04 | 0.42* | 0.53* | 0.03 | 0.42* | 0.54* | 0.03 | 0.01* | 0.04* | 0.02 | 0.06* | 0.12* | 0.02 |
| 0.05 | 4.91 | 0.04 | 0.04 | 0.22 | 0.03 | 0.04 | 0.23 | 0.03 | 0.04 | 0.24 | 0.02* | 0.12* | 0.08 | 0.21* | 0.46* | 0.08 |
| 1.0 | 9.62 | 0.29* | 0.33* | 0.02 | 0.21* | 0.27* | 0.02 | 0.21* | 0.27* | 0.02 | 0.01* | 0.03* | 0.01 | 0.04* | 0.07* | 0.01 |
| 2.0 | 11.51 | 0.07 | 0.08 | 0.44 | 0.06 | 0.08 | 0.49 | 0.07 | 0.08 | 0.47 | 0.02* | 0.19* | 0.22 | 0.21* | 0.47* | 0.21 |
| 3.0 | 11.51 | 0.12 | 0.15 | 0.84 | 0.12 | 0.15 | 0.94 | 0.12 | 0.16 | 0.95 | 0.10* | 0.87* | 0.46 | 0.29* | 0.97* | 0.46 |
| 4.0 | 14.09 | 0.12 | 0.15 | 0.84 | 0.12 | 0.15 | 0.94 | 0.12 | 0.16 | 0.95 | 0.10* | 0.87* | 0.46 | 0.29* | 0.97* | 0.46 |
| 5.0 | 14.09 | 0.23 | 0.27 | 1.67 | 0.23 | 0.30 | 1.85 | 0.22 | 0.28 | 1.82 | 0.18* | 0.01 | 0.87 | 0.36* | 0.02 | 0.86 |
| 7.0 | 14.09 | 0.23 | 0.27 | 1.67 | 0.23 | 0.30 | 1.85 | 0.22 | 0.28 | 1.82 | 0.18* | 0.01 | 0.87 | 0.36* | 0.02 | 0.86 |
| 10.0 | 17.75 | 0.23 | 0.27 | 1.67 | 0.23 | 0.30 | 1.85 | 0.22 | 0.28 | 1.82 | 0.18* | 0.01 | 0.87 | 0.36* | 0.02 | 0.86 |
| 15.0 | 17.75 | 0.43 | 0.50 | 3.20 | 0.42 | 0.56 | 3.79 | 0.38 | 0.49 | 3.66 | 0.62* | 0.05 | 1.67 | 0.81* | 0.06 | 1.67 |
| 20.0 | 23.09 | 0.43 | 0.50 | 3.20 | 0.42 | 0.56 | 3.79 | 0.38 | 0.49 | 3.66 | 0.62* | 0.05 | 1.67 | 0.81* | 0.06 | 1.67 |
| 25.0 | 23.09 | 0.76 | 0.89 | 6.24 | 0.77 | 1.04 | 7.39 | 0.63 | 0.81 | 6.80 | 0.50* | 0.05 | 3.81 | 0.69* | 0.05 | 3.81 |
| 50.0 | 31.02 | 0.76 | 0.89 | 6.24 | 0.77 | 1.04 | 7.39 | 0.63 | 0.81 | 6.80 | 0.50* | 0.05 | 3.81 | 0.69* | 0.05 | 3.81 |
| 100.0 | 43.29 | 1.40 | 1.61 | 12.32 | 1.40 | 1.88 | 14.59 | 1.04 | 1.34 | 12.95 | 0.98* | 0.11 | 7.52 | 0.01 | 0.11 | 7.52 |
| 150.0 | 62.72 | 2.44 | 2.84 | 27.58 | 2.48 | 3.32 | 27.63 | 1.77 | 2.28 | 23.97 | 0.03 | 0.35 | 19.50 | 0.03 | 0.35 | 19.50 |
| 200.0 | 62.72 | 4.26 | 4.98 | 47.65 | 4.28 | 5.72 | 57.44 | 3.07 | 3.98 | 44.00 | 0.06 | 0.57 | 28.00 | 0.06 | 0.57 | 28.00 |

\* Values scaled by 100

**Table B6.** Error values and compression ratios of the `ZFP` baseline run on the `tra_20000101_00_permuted` dataset. Note that for some different tolerances the compression ratios are the same, this appears to be related to how `ZFP` works.

| Tolerance | Ratio | lon-lat | | | p | | | temperature | | | potential vorticity | | | specific humidity | | |
|---|---|---|---|---|---|---|---|---|---|---|---|---|---|---|---|---|
| | | L1 | RMSE | L-infinity | L1 | RMSE | L-infinity | L1 | RMSE | L-infinity | L1 | RMSE | L-infinity | L1 | RMSE | L-infinity |
| 0.1 | 3.99 | 0.58 | 0.67 | 0.03 | 0.42 | 0.54 | 0.03 | 0.42 | 0.54 | 0.03 | 0.02 | 0.08 | 0.04 | 0.14 | 0.22 | 0.05 |
| 0.5 | 5.47 | 0.04 | 0.05 | 0.27 | 0.03 | 0.04 | 0.24 | 0.03 | 0.04 | 0.24 | 0.08 | 0.49 | 0.10 | 0.26 | 0.65 | 0.10 |
| 0.05 | 3.64 | 0.30 | 0.34 | 0.02 | 0.21 | 0.27 | 0.01 | 0.21 | 0.27 | 0.02 | 0.02 | 0.06 | 0.02 | 0.08 | 0.12 | 0.02 |
| 1.0 | 6.17 | 0.07 | 0.09 | 0.49 | 0.07 | 0.09 | 0.48 | 0.07 | 0.09 | 0.47 | 0.13 | 0.01 | 0.22 | 0.32 | 0.01 | 0.21 |
| 2.0 | 7.05 | 0.14 | 0.17 | 0.93 | 0.13 | 0.17 | 0.97 | 0.13 | 0.17 | 0.95 | 0.33 | 0.03 | 0.55 | 0.52 | 0.03 | 0.55 |
| 3.0 | 7.05 | 0.14 | 0.17 | 0.93 | 0.13 | 0.17 | 0.97 | 0.13 | 0.17 | 0.95 | 0.33 | 0.03 | 0.55 | 0.52 | 0.03 | 0.55 |
| 4.0 | 8.18 | 0.28 | 0.33 | 1.78 | 0.26 | 0.33 | 1.99 | 0.25 | 0.31 | 1.84 | 0.56 | 0.05 | 0.87 | 0.75 | 0.05 | 0.86 |
| 5.0 | 8.18 | 0.28 | 0.33 | 1.78 | 0.26 | 0.33 | 1.99 | 0.25 | 0.31 | 1.84 | 0.56 | 0.05 | 0.87 | 0.75 | 0.05 | 0.86 |
| 7.0 | 8.18 | 0.28 | 0.33 | 1.78 | 0.26 | 0.33 | 1.99 | 0.25 | 0.31 | 1.84 | 0.56 | 0.05 | 0.87 | 0.75 | 0.05 | 0.86 |
| 10.0 | 9.66 | 0.52 | 0.61 | 3.44 | 0.51 | 0.64 | 3.82 | 0.44 | 0.56 | 3.79 | 0.90 | 0.08 | 1.67 | 0.01 | 0.08 | 1.68 |
| 15.0 | 9.66 | 0.52 | 0.61 | 3.44 | 0.51 | 0.64 | 3.82 | 0.44 | 0.56 | 3.79 | 0.90 | 0.08 | 1.67 | 0.01 | 0.08 | 1.68 |
| 20.0 | 11.52 | 0.95 | 1.11 | 6.80 | 0.95 | 1.20 | 7.63 | 0.74 | 0.95 | 7.01 | 0.02 | 0.19 | 4.57 | 0.02 | 0.19 | 4.56 |
| 25.0 | 11.52 | 0.95 | 1.11 | 6.80 | 0.95 | 1.20 | 7.63 | 0.74 | 0.95 | 7.01 | 0.02 | 0.19 | 4.57 | 0.02 | 0.19 | 4.56 |
| 50.0 | 13.84 | 1.73 | 1.99 | 13.42 | 1.71 | 2.20 | 14.91 | 1.25 | 1.59 | 14.32 | 0.04 | 0.36 | 7.52 | 0.04 | 0.36 | 7.52 |
| 100.0 | 16.73 | 3.25 | 3.71 | 28.29 | 3.02 | 3.90 | 28.94 | 2.31 | 2.90 | 27.03 | 0.10 | 0.84 | 19.51 | 0.10 | 0.84 | 19.50 |
| 150.0 | 20.67 | 6.33 | 7.23 | 49.24 | 5.24 | 6.71 | 54.17 | 4.51 | 5.63 | 52.39 | 0.16 | 1.37 | 28.00 | 0.17 | 1.37 | 28.00 |
| 200.0 | 20.67 | 6.33 | 7.23 | 49.24 | 5.24 | 6.71 | 54.17 | 4.51 | 5.63 | 52.39 | 0.16 | 1.37 | 28.00 | 0.17 | 1.37 | 28.00 |

\* Values scaled by 100



**Table B7.** Jensen-Shannon divergence metrics for different time steps for trajectories starting at $1\,\mathrm{hPa}$

| Dataset | 2D | | | Pressure | | |
|---|---|---|---|---|---|---|
| | 72h | 120h | 168h | 72h | 120h | 168h |
| Noise 0.01 | 9.94e-04 | 1.73e-03 | 2.53e-03 | 3.10e-04 | 8.43e-03 | 3.28e-04 |
| Noise 0.05 | 4.46e-03 | 5.87e-03 | 7.03e-03 | 1.07e-01 | 1.10e-01 | 1.19e-02 |
| Noise 0.1 | 7.15e-03 | 9.38e-03 | 1.03e-02 | 2.81e-01 | 1.09e-01 | 4.97e-02 |
| Compressed 1.67x | 3.84e-06 | 3.30e-06 | 3.85e-06 | 1.99e-07 | 9.15e-08 | 1.71e-07 |
| Compressed 12.8x | 4.23e-04 | 1.01e-03 | 1.97e-03 | 1.17e-04 | 1.90e-04 | 1.26e-04 |
| Compressed 44.5x | 3.10e-03 | 7.47e-03 | 1.31e-02 | 6.93e-04 | 1.50e-03 | 1.40e-03 |
| Wind Bitrounding | 1.26e-01 | 1.39e-01 | 1.56e-01 | 6.31e-01 | 6.41e-01 | 6.02e-01 |

**Table B8.** Jensen-Shannon divergence metrics for different time steps for trajectories starting at $500\,\mathrm{hPa}$

| Dataset | 2D | | | Pressure | | |
|---|---|---|---|---|---|---|
| | 72h | 120h | 168h | 72h | 120h | 168h |
| Noise 0.01 | 5.51e-05 | 2.84e-04 | 9.68e-04 | 4.56e-02 | 3.36e-01 | 1.03e-01 |
| Noise 0.05 | 2.43e-04 | 1.02e-03 | 2.33e-03 | 4.59e-02 | 3.36e-01 | 1.01e-01 |
| Noise 0.1 | 4.42e-04 | 1.59e-03 | 3.34e-03 | 4.51e-02 | 3.28e-01 | 9.66e-02 |
| Compressed 1.67x | 1.96e-06 | 1.16e-06 | 1.10e-06 | 4.56e-02 | 3.35e-01 | 1.03e-01 |
| Compressed 12.8x | 8.61e-04 | 1.45e-03 | 1.82e-03 | 4.27e-02 | 3.36e-01 | 9.93e-02 |
| Compressed 44.5x | 6.57e-03 | 1.05e-02 | 1.23e-02 | 3.27e-02 | 3.16e-01 | 8.84e-02 |
| Wind Bitrounding | 1.85e-01 | 2.66e-01 | 3.17e-01 | 1.41e-01 | 5.14e-01 | 2.46e-01 |

**Table B9.** Jensen-Shannon divergence metrics for different time steps for trajectories starting at $1000\,\mathrm{hPa}$

| Dataset | 2D | | | Pressure | | |
|---|---|---|---|---|---|---|
| | 72h | 120h | 168h | 72h | 120h | 168h |
| Noise 0.01 | 7.80e-05 | 5.40e-04 | 1.64e-03 | 3.80e-01 | 1.73e-01 | 2.06e-01 |
| Noise 0.05 | 3.45e-04 | 1.51e-03 | 3.23e-03 | 3.74e-01 | 1.67e-01 | 2.05e-01 |
| Noise 0.1 | 5.82e-04 | 2.21e-03 | 4.24e-03 | 3.74e-01 | 1.72e-01 | 2.06e-01 |
| Compressed 1.67x | 2.24e-06 | 1.60e-06 | 2.83e-06 | 3.80e-01 | 1.73e-01 | 2.06e-01 |
| Compressed 12.8x | 7.22e-04 | 1.14e-03 | 1.64e-03 | 3.79e-01 | 1.67e-01 | 2.04e-01 |
| Compressed 44.5x | 5.53e-03 | 1.01e-02 | 1.25e-02 | 3.62e-01 | 1.54e-01 | 1.96e-01 |
| Wind Bitrounding | 4.22e-02 | 8.53e-02 | 1.24e-01 | 1.75e-01 | 1.40e-01 | 3.87e-01 |





**Appendix C: User Manual**

In this section a small user manual for psit is given. For a detailed description please refer to (Pietak, 2025a). The psit compression pipeline has been implemented in Python, as this allowed for fast development and quick iteration of different ideas. The entire compression and decompression logic has been implemented in a class called `Psit`, which exposes two functions: `compress()` and `decompress()`. The `compress()` function takes an `xarray.Dataset` (with additional configu-

570 ration parameters) as input and compresses it to a file on disk. The `decompress()` function takes the location of such a compressed file and returns an `xarray.Dataset` with the decompressed trajectory data. The additional configuration parameters, which can be passed to the `compress()` function, and a brief description are provided below.

**dataset** The `xarray.Dataset` we wish to compress.

**filename** The file in which the compressed data should be stored.

**crf** The compression factor.

**exclude** List of data variables which should not be compressed.

**method** The compression algorithm that should be used.

**color_method** The color encoding method that should be used.

**color_bits** The number of bits to use per color channel (when using color encoding).

**delta_method** The delta encoding method that should be used.

**local** If the trajectory starting positions are in a local domain and not global.

**bin** The number of pressure bins that should be used.

**mapping_func** Which mapping method should be used.

**num_workers** The number of parallel workers that should be used.

**factor** The ratio between number of grid points and trajectories.

**lon_name** Name of the longitude variable in the dataset.

**lat_name** Name of the latitude variable in the dataset.

**p_name** Name of the pressure variable in the dataset.

Note that some of these parameters, like `crf` or `method`, allow the individual setting of the parameter for each of the data

variables available (using a dictionary mapping from variable name to value) in the dataset.





In addition to the `Psit` class, a CLI wrapper has been written called `psitcli`. This CLI wrapper allows for the easy compression of netCDF (Unidata, 2024) files from the command line. The program can be run with the command `python psitcli.py {compress/decompress/test} [options] in_file out_file`. The first option that needs to be defined is the mode to be used, this can either be `compression`, `decompression`, or `test`. The first two are used for compression and decompression, while the `test` mode will perform a compression followed by a decompression and the calculation of performance metrics such as the L1 error that was created. These additional metrics are stored as a Python `pickle` dump in a file called `<out_file>_data.pick`. The `compress` and `test` modes both take additional options, which mirror the ones accepted by the `compress()` function of the `Psit` class. Alternatively, these options can be defined via a YAML configuration file, which is then passed to `psitcli`. An example of such a configuration file is given in Listing 1.

*Author contributions.* AP, LH, and LF designed psit. AP implemented and ran the experiments using psit. MS and SS provided domain specific knowledge and general advice. TH supervised the project. AP prepared the manuscript with the help of all other co-authors.

*Competing interests.* The authors declare that they have no conflict of interest.

*Acknowledgements.* We would like to thank CSCS the Swiss supercomputer center for providing us with hardware resources without which this project could not have been done.

AP would like to thank James Murdoch MacGregor (J. T. McIntosh), for writing the novelet "Humanoid Sacrifice" (McIntosh, 1964) and creating the name "psit".





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
