# Peer review of "psit 1.0: A System to Compress Lagrangian Flows"

_EGUsphere, 2025_

## Author Comment (AC1)

**Response to Dr. Underwood**

Alexander Pietak, Langwen Huang, Luigi Fusco, Michael Sprenger, Sebastian Schemm, and Torsten Hoefler

Dear Dr. Underwood. Thank you for taking the time to provide a review of our submitted manuscript, submitted on 14 Mar 2025. We are pleased your review is supportive of our work and we are of course happy to response to points raised and revise the manuscript accordingly.

The primary improvements we have implemented is the addition of SZ3 as a second baseline comparison to the experiments, along with the inclusion of a rate distortion plot. Additionally, please find below more detailed responses to all your comments.

*Comment 1): On line 336, the authors choose a "compression factor of 30". Why was this value chosen? Additionally, the authors should consider multiple values of CRs for this analysis.*

**Response:** Thank you for your suggestion. We have made significant revisions to this experiment, during which we now compare the error distribution of psit with that of ZFP and SZ3 for two different compression ratios of approximately 2.5 and 15. By providing these two compression ratios, we offer a more comprehensive overview of the performance. You may find updated version of the plots in Fig. 1.

*Comment 2): This paper misses key references to work on the compression of unstructured grid data which would be more appropriate comparisons for this work*

**Response:** Thank you for the suggestion; we have added a new paragraph in the introduction section where we discuss these papers. Not a large focus is given on these algorithms as they are not inherently designed to compress trajectory data.

*Comment 3): The paper should reference the notion of rate distortion in the discussion of Figure 12, and include the PSNR to be more consistent with the standard presentation of rate distortion results, as done in the ZFP and SZ line of papers*

**Response:** Thank you for bringing our attention to this. We have revisited this experiment and have added a PSNR plot. You may find an excerpt of this new analysis in the form of the PSNR plot of the central angle error in Fig. 2.

*Comment 4): Some statements in the paper are not insightful, such as "In general, the L-infinity error is larger than the L1 and RMSE errors." By definition, it is always true that $L_\infty \geq L_1$ because the max upper bounds the average.Some statements in the paper are not insightful, such as "In general, the L-infinity error is larger than the L1 and RMSE errors." By definition, it is always true that $L_\infty \geq L_1$ because the max upper bounds the average.*

**Response:** We have removed the statement and rewritten parts of the analysis for several experiments.

*Comment 5): What justifies the author's "[exception that] the error distribution ... [follows] a Gaussian". This actually depends on the design of the compressor and the error bound chosen. See "Error Distributions of Lossy Floating-Point Compressors" by Lindstrom 2017.*

**Response:** We have reworked this experiment. Our intent is to emphasize that most errors are concentrated at 0.

[Figure]

**Figure 1.** Error distribution based on values for the `tra_20200101_00_permuted` dataset. We compress with compression ratios of 2.5, and 15. The area of the densities have been normalized and the x axis is cropped to 5 standard deviations of the psit distribution for all cases, except pressure with 2.5 times compression where it is 1 standard deviation. The data variables are pressure (p), potential temperature (TH), and potential vorticity (PV).

[Figure]

**Figure 2.** PSNR against compression ratio for the central angle error of the `tra_20200101_00` and the `tra_20200101_00_permuted` trajectory files. The shaded regions for ZFP and SZ3 corresponds to the range in compression performance between the different files.

[Figure]

**Figure 3.** Excerpt of the comparison between psit, ZFP, and SZ3 for the `tra_20200101_00` and `tra_20200101_00_permuted` files. The RMSE and L-infinity error is compared to the achieved compression ratio for the central angle error (lon-lat). In the plots the shaded area for ZFP and SZ3 corresponds to the range in compression performance between the two files. Note that psit performs the same for both, therefore, only one line is plotted.

*Comment 6): The limitations of bit rounding-based compressors are well documented in prior work for both SZ and ZFP-based compressors, including papers cited here. It is unclear why this was used as a comparison in section 3.3 of the paper.*

**Response:** Thank you for highlighting this issue. As bitrounding is not the major focus of this experiment and distracted from the comparison, we have taken it out of the manuscript.

*Comment 7): It is not clear why the authors compared ZFP and JPEG and not SZ3 when they use ZFP as part of their pipeline in some configurations. It is well known that SZ3 tends to get much higher compression ratios at each quality threshold.*

**Response:** Sorry for the confusion. We have incorporated SZ3 as a second baseline comparison in the experiments, where we are currently only comparing to ZFP. You may find an excerpt of this additional analysis in Fig. 3.

*Comment 8): There is extensive work on quantifying appropriate error thresholds by Millian Klöwer and Alison Baker using metrics such as SSIM and dSSIM. Why did the authors not use the metrics in their analysis?*

**Response:** Thank you for your suggestion. We did not include SSIM and dSSIM in the manuscript as they are primarily designed for images (i.e., grid data), but we are focusing on trajectories.

*Comment 9): The authors should specify the versions of ZFP and SZ3 used in their work, as newer versions have updated the default algorithm to higher-performing versions. For ZFP, there are versions with much higher parallel performance or support for additional modes with parallel compression.*

**Response:** We have included the versions in the manuscript. For ZFP, it would be pyzfp v0.5.5, which wraps ZFP version 0.5.5, and for SZ3, it is version 3.2.1.

*Comment 10): The authors should specify the error bounding type (runtime) and rounding mode (compile time) used with ZFP for their results.*

**Response:** We have added the corresponding information to the manuscript. The bounding type is Fixed-Accuracy. The ZFP version used does not support rounding modes.

*Comment 11): E.g., "While this approach is conceptually sound, it quickly, [sic] becomes computationally infeasible, so it found little use in the finished pipeline. However, we still include it here for its theoretical insights." could have been written*

[Figure]

**Figure 4.** Latitude grids with $550\,\mathrm{hPa}$ starting pressure created with the bipartite mapping method, for detailed information on this refer to section 2.2 of the manuscript. Three different time steps are displayed the leftmost image is at the first time step, the second one is 12 hours later and the rightmost one is 24 hours later. In the leftmost image we have very good smoothness, which originates from the mapping method, over time this smoothness starts to degrade, as the trajectories start to diverge.

*"solving an LP is computational infeasible for large images, but included for theoretical analysis."*

**Response:** We have incorporated it into the manuscript.

55 *Comment 12): Additionally, there are many run-on sentences that span 3 or more lines of text.*

**Response:** Thank you for pointing out. We have fixed this.

*Comment 13): Lastly, 21 figures (not including subfigures) seems excessive*

**Response:** We have distilled our figures down to the important ones and are now at 13 figures in the main part.

*Question 1): On line 330, the authors state, "For longer time ranges, the performance of psit starts to degrade.", Do the authors*

60 *have an explanation that explains this discrepancy?*

**Response:** The reason is, over time, trajectories start to diverge, which results in the smoothness of the images — dependent on their initial position — being lost. This can be seen in Fig. 4 which corresponds to figure 4 of the manuscript.

---

## Author Comment (AC2)

**Response to Anonymous Referee 2**

Alexander Pietak, Langwen Huang, Luigi Fusco, Michael Sprenger, Sebastian Schemm, and Torsten Hoefler

Dear Colleague. Thank you for taking the time to provide a review of our submitted manuscript, submitted on 14 Mar 2025. We are pleased your review is supportive of our work and we are of course happy to response to points raised and revise the manuscript accordingly. Please find below answers to each of the comments you have made.

5 *General Comment 1): I'd like to see more justification/explanation for different experimental choices. It's not clear to me why certain compressor choices were made for different experiments. For example in 3.2., the text says that Listing 1 led to "good results". Based on what? Was it optimized somehow? Then bitrounding is used for one variable later on. Also why is the comparison between psit and zfp? Can sz3 only used within psit? Can zfp not be used within psit as well on the grids? I would think zfp could be more effective than jpeg ...*

10 **Response:** Thanks for pointing out, we have made the following adjustments in order to improve clarity and answer your questions:

- For the configuration of psit we added an ablation study, of which the main findings can be found in Table 1.

- We originally added bitrounding to have another comparison in the perturbed wind fields section, but we have removed it as it is not the major focus of this experiment.

15 - Currently psit only supports either JPEG 2000 and SZ3, in the future it could be expanded to also support ZFP.

- We now added SZ3 as another comparison in the experiments.

*General Comment 2): It is not always clear why a specific amount of compression was chosen.*
**Response:** Thank you for pointing this out. In the experiments we present a range of different compression ratios in order for the reader to gain an overview of how psit behaves in different situations. We improved the manuscript in order to make these
20 intentions more clear.

*General Comment 3): Many of the figure and table captions could be improved with more text / explanations.*
**Response:** We went though all the figures and improve their respective captions.

*Comment 1): Line 286: Why is sz3 used for pressure? Did other approaches not work for this variable? Were other approaches superior to sz3 on the other variables?*
25 **Response:** Our experiments showed that JPEG 2000 generally leads to lower errors except for pressure where it was SZ3. Our results indicate that this has to do with the overall lack of smoothness in the pressure variable.

**Table 1.** Ablation study for psit on the `tra_20200101_00` trajectory file. For different configurations the file is compressed to a compression ratio of around 20 after which the RMSE error for the different data variables is considered. If we then pick the lowest value for each error column we can figure out the optimal configuration. It consists of using JPEG 2000 with delta encoding for all data variables except pressure where SZ3 with no delta encoding should be used, additionally we should use XYZ color encoding.

| Configuration | RMSE | | | |
| --- | --- | --- | --- | --- |
| | **lon-lat** | **p** | **TH** | **PV** |
| JPEG 200, no color, no delta | 0.283 | 3.45 | 0.270 | 0.0942 |
| SZ3, no color, no delta | 0.613 | **1.59** | 0.256 | 0.116 |
| JPEG 200, no color, delta | 0.161 | 2.12 | **0.170** | **0.0768** |
| SZ3, no color, delta | 2.38 | 1.95 | 0.197 | 0.162 |
| JPEG 200, XYZ color, delta | **0.118** | – | – | – |

*Comment 2): Line 337: Not all lossy compressors result in Gaussian error distributions. Please see Peter Lindstrom's paper from 2017 (https://www.osti.gov/servlets/purl/1526183)*

**Response:** Thanks for pointing this out. We have reworked this experiment, by also including other compression algorithms. Our goal is to emphasize that most errors are concentrated at 0. You may find an updated version of the distribution plot in Fig. 1.

*Comment 3): Line 336: Why did you choose a factor of 30?*

**Response:** We chose 30 as it struck a middle ground in the compression range that we explored. We now examine compression ratios of 2.5 and 15 in order to diversify this.

*Comment 4): Line 351: Why did you choose to use bitrounding on the wind field? Bitrounding was not mentioned previously, so it feels like a surprise.*

**Response:** Thanks for pointing this out. As bitrounding was not the major focus of this experiment and distracted from the comparison, we have taken it out of the manuscript.

*Comment 5): Line 374-375: For bitrounding, the lossless compressor that it is paired with can make a big difference (e.g., the newish Pcodec, or Pco, compressor can be quite an improvement). Which lossless method was used here?*

**Response:** Sorry for the confusion. For bitrounding no lossless backend was used, as we do not use it as a compression baseline, but as a method for creating perturbed wind fields. We take the original wind fields and apply bitrounding to them, after this we trace trajectories on these bitrounded wind fields and compared them to trajectories (starting at the same locations traced on the original wind field) which have been compressed by psit. We have removed bitrouding from this experiment as is was not a major focus and distracted from the comparison.

*Comment 6): What versions of zfp and sz3 did you use? For zfp, it looks like you used the absolute tolerance mode (hence the "tolerance" in the tables in the appendix), but this is not specified.*

[Figure]

**Figure 1.** Error distribution based on values for the `tra_20200101_00_permuted` dataset. We compress with compression ratios of 2.5, and 15. The area of the densities have been normalized and the x axis is cropped to 5 standard deviations of the psit distribution for all cases, except pressure with 2.5 times compression where it is 1 standard deviation. The data variables are pressure (p), potential temperature (TH), and potential vorticity (PV).

**Response:** We have added this information to the manuscript. For ZFP it is pyzfp v0.5.5 (which wraps ZFP v0.5.5) and here we are using the fixed-accuracy (tolerance) mode. For SZ3 we are using version 3.2.1 in the relative error mode.

*Comment 7): Line 441, re: "..the input data trajectories need to be uniformly distributed." : How common is this in practice? I don't have a sense on whether this requirement is restrictive or not.*

**Response:** Thank you for pointing out. We have updated the manuscript with references to related work in which showcase different initial distributions. Examples for global uniform distributions would be (Stoffels et al., 2025; Sprenger et al., 2017; Bakels et al., 2025). Examples for local uniform distributions would be (Pérez-Muñuzuri et al., 2018; Wendisch et al., 2024). A third type of initial condition (which are outside of psit's scope) are non uniform distributions examples for this are (Keune et al., 2022; Schielicke and Pfahl, 2022; Dey et al., 2023).

*Comment 8): Section 4.2 (3rd paragraph): There is a lot of existing work that argues that simple metrics are not sufficient for evaluating the effects of lossy compression on weather and climate data. At least some other work should be cited. Here are a few earlier works that come to mind:*

- *Baker 2014: doi:10.1145/2600212.2600217*

- *Baker 2016: doi:10.5194/gmd-9-4381-2016POPPICK2020104599*

**Table 2.** The minimal and maximal values for the different data variables of the `tra_20200101_00` file.

| Data variable | Minimum | Maximum |
|---|---|---|
| Pressure | 32 hPa | 974 hPa |
| Potential temperature | 256 K | 559 K |
| Potential vorticity | −980 PVU | 66 PVU |

**Table 3.** The minimal and maximal values for the different data variables of the `tra_20200101_00` file.

| Data variable | Minimum | Maximum |
|---|---|---|
| Pressure | 0 hPa | 1500 hPa |
| Temperature | 182 K | 313 K |
| Potential vorticity | $-21 \, \mathrm{K m^2 kg^{-1} s^{-1}}$ | $30 \, \mathrm{K m^2 kg^{-1} s^{-1}}$ |
| Specific humidity | $0 \, \mathrm{kg kg^{-1}}$ | $30 \, \mathrm{kg kg^{-1}}$ |

  – *Poppick 2020: doi:10.1016/j.cageo.2020.104599*

**Response:** Thank you for pointing this out, we have now integrated this into the manuscript during the introduction of Section 3.

*Comment 9): Tables B1-B6: I don't particularly care for the choice to scale some of the values by 100 (indicated by a "*'). It makes it harder to glance down the column. Maybe use more digits or round or ?*

**Response:** Thanks for pointing this out. We have switched to a scientific notation style in the tables, which makes them more easily readable.

*Comment 10): Table B2-B3: "appears to be related to how ZFP works". Can you provide a more meaningful explanation?*

**Response:** This arises from the grouping into the different "bit planes" that is done by the embedded coding strategy of ZFP. Different tolerances lead to the same grouping and therefore to the same compression behaviour. We have added this explanation to the manuscript.

*Comment 11): FIgure 12, 13: Have you considered normalizing these error metrics for plotting so that the y-axis extents could potentially be the same for each error metric across a variable?*

**Response:** We have added normalized versions of the errors as well as PSNR plots to the manuscript. You may find the normalized error plots for the `tra_20200101_00` dataset in Fig. 2.

*Comment 12): I'd like to know what the min/max values are for the variables that are being compressed so I have an idea of what an error tolerance of 20 means, for example.*

**Response:** We have added two tables which display the minimum and the maximum for the different variables of the two trajectory files we consider. You may find them in Tables 2 and 3.

[Figure]

**Figure 2.** Data variable normalized error comparison between psit, ZFP, and SZ3 for the `tra_20200101_00` and `tra_20200101_00_permuted` files. The L1, RMSE, and L-infinity errors are normalized and compared to the achieved compression ratio for the central angle error (lon-lat) and the other data variables (pressure (p), potential temperature (TH), and potential vorticity (PV)). In the plots the shaded area for ZFP and SZ3 corresponds to the range in compression performance between the two files. Note that psit performs the same for both, therefore, only one line is plotted.

*Comment 13): Appendix data. There is a lot of data in the appendices (especially B) , which isn't necessarily a problem, but it should be there for a reason (i.e., referred to with some discussion in the paper or appendix itself). And the volume makes it harder to make meaningful comparisons. For example, for tables B1 and B2, what is interesting to me is to compare the error values for psit versus zfp at factors/tolerances which yield a similar compression ratio. So, listing psit with factor 25 next to zfp with tolerance 2.0 is informative because both yield a compression rate of 10. That helps me consider which is better quality-wise for the same data reduction. Also I'm skeptical that the amount if compression in the lower rows of these tables (e.g., B1 and B2) is something that would ever be used in practice for climate and weather data, but feel free to argue otherwise.*

**Response:** Thank you for bringing up this concern. We made sure that every item in the appendix gets referenced in the text. Our goal is that we have the plots which can be used to make comparisons between the different compression methods and the tables come into play if one is interested in the exact values. We include the smaller compression ratios in order to provide a broader range of compression scenarios.

*Minor Issue 1): Consider using the same block font for psit that you have used for zfp, sz3, jpeg2000, etc.*

**Response:** We have switched from the block font to a normal one for all mentioned names.

95    *Minor Issue 2): Line 250: Awkward phrasing*

**Response:** Thanks for pointing this out. We rewrote the mentioned section also incorporating the point you have brought up in comment 8.

*Minor Issue 3): Line 251: "ZFP" is in a block font in other occurrences in the paper*

**Response:** We have resolved this.

100   *Minor Issue 4): Most (if not all) opening quotes appear to be backwards (e.g., line 118)*

**Response:** We have fixed the backward opening quotes.

*Minor Issue 5): line 156: "denotes if" => "denotes whether"*

**Response:** We have replaced it.

*Minor Issue 6): Figure 5: consider making this caption more descriptive*

105   **Response:** We have reworked the description to describe the process in the figure more detailed.

*Minor Issue 7): Figure 16-18: color bars are not labeled*

**Response:** We have added the label "Density".

**References**

Bakels, L., Blaschek, M., Dütsch, M., Plach, A., Lechner, V., Brack, G., Haimberger, L., and Stohl, A.: LARA: a Lagrangian Reanalysis based on ERA5 spanning from 1940 to 2023, Earth System Science Data, 17, 4569–4585, https://doi.org/10.5194/essd-17-4569-2025, https://essd.copernicus.org/articles/17/4569/2025/, 2025.

Dey, D., Aldama Campino, A., and Döös, K.: Atmospheric water transport connectivity within and between ocean basins and land, Hydrology and Earth System Sciences, 27, 481–493, https://doi.org/10.5194/hess-27-481-2023, https://hess.copernicus.org/articles/27/481/2023/, 2023.

Keune, J., Schumacher, D. L., and Miralles, D. G.: A unified framework to estimate the origins of atmospheric moisture and heat using Lagrangian models, Geoscientific Model Development, 15, 1875–1898, https://doi.org/10.5194/gmd-15-1875-2022, https://gmd.copernicus.org/articles/15/1875/2022/, 2022.

Pérez-Muñuzuri, V., Eiras-Barca, J., and Garaboa-Paz, D.: Tagging moisture sources with Lagrangian and inertial tracers: application to intense atmospheric river events, Earth System Dynamics, 9, 785–795, https://doi.org/10.5194/esd-9-785-2018, https://esd.copernicus.org/articles/9/785/2018/, 2018.

Schielicke, L. and Pfahl, S.: European heatwaves in present and future climate simulations: a Lagrangian analysis, Weather and Climate Dynamics, 3, 1439–1459, https://doi.org/10.5194/wcd-3-1439-2022, https://wcd.copernicus.org/articles/3/1439/2022/, 2022.

Sprenger, M., Fragkoulidis, G., Binder, H., Croci-Maspoli, M., Graf, P., Grams, C. M., Knippertz, P., Madonna, E., Schemm, S., Škerlak, B., and Wernli, H.: Global Climatologies of Eulerian and Lagrangian Flow Features based on ERA-Interim, Bulletin of the American Meteorological Society, 98, 1739 – 1748, https://doi.org/10.1175/BAMS-D-15-00299.1, https://journals.ametsoc.org/view/journals/bams/98/8/bams-d-15-00299.1.xml, 2017.

Stoffels, R., Benedict, I., Papritz, L., Selten, F., and Weijenborg, C.: Precipitation, Moisture Sources and Transport Pathways associated with Summertime North Atlantic Deep Cyclones, EGUsphere, 2025, 1–36, https://doi.org/10.5194/egusphere-2025-1752, https://egusphere.copernicus.org/preprints/2025/egusphere-2025-1752/, 2025.

Wendisch, M., Crewell, S., Ehrlich, A., Herber, A., Kirbus, B., Lüpkes, C., Mech, M., Abel, S. J., Akansu, E. F., Ament, F., Aubry, C., Becker, S., Borrmann, S., Bozem, H., Brückner, M., Clemen, H.-C., Dahlke, S., Dekoutsidis, G., Delanoë, J., De La Torre Castro, E., Dorff, H., Dupuy, R., Eppers, O., Ewald, F., George, G., Gorodetskaya, I. V., Grawe, S., Groß, S., Hartmann, J., Henning, S., Hirsch, L., Jäkel, E., Joppe, P., Jourdan, O., Jurányi, Z., Karalis, M., Kellermann, M., Klingebiel, M., Lonardi, M., Lucke, J., Luebke, A. E., Maahn, M., Maherndl, N., Maturilli, M., Mayer, B., Mayer, J., Mertes, S., Michaelis, J., Michalkov, M., Mioche, G., Moser, M., Müller, H., Neggers, R., Ori, D., Paul, D., Paulus, F. M., Pilz, C., Pithan, F., Pöhlker, M., Pörtge, V., Ringel, M., Risse, N., Roberts, G. C., Rosenburg, S., Röttenbacher, J., Rückert, J., Schäfer, M., Schaefer, J., Schemann, V., Schirmacher, I., Schmidt, J., Schmidt, S., Schneider, J., Schnitt, S., Schwarz, A., Siebert, H., Sodemann, H., Sperzel, T., Spreen, G., Stevens, B., Stratmann, F., Svensson, G., Tatzelt, C., Tuch, T., Vihma, T., Voigt, C., Volkmer, L., Walbröl, A., Weber, A., Wehner, B., Wetzel, B., Wirth, M., and Zinner, T.: Overview: quasi-Lagrangian observations of Arctic air mass transformations – introduction and initial results of the HALO–$(\mathcal{AC})^3$ aircraft campaign, Atmospheric Chemistry and Physics, 24, 8865–8892, https://doi.org/10.5194/acp-24-8865-2024, https://acp.copernicus.org/articles/24/8865/2024/, 2024.